# L-2-hydroxyglutarate regulates centromere and heterochromatin conformation in the male germline

Nina Mayorek[1], Miriam Schlossberg[1], Yousef Mansour[1], Nir Pillar [2], Ilan Stein[1,2],
Fatima Mushasha[1], Guy Baziza Paz[1], Eleonora Medvedev[3], Zakhariya Manevitch[3],
Julia Menzel[4], Elina Aizenshtein[5], Boris Sarvin[5,6], Nikita Sarvin[5,6], Erwin Goldberg[7]
Bryan A. Niedenberger[8], Christopher B. Geyer[8,9], Tomer Shlomi[5,6,10], Michael Klutstein[11],
Eli Pikarsky [1,2]*

1 The Concern Foundation Laboratories at The Lautenberg Center for Immunology and Cancer Research,
Israel-Canada Medical Research Institute, Faculty of Medicine, The Hebrew University, Jerusalem, Israel,
2 Department of Pathology, Hadassah-Hebrew University Medical Center, Jerusalem, Israel, 3 The Core
Research Facility, Hebrew University-Hadassah Medical School, Jerusalem, Israel, 4 Abberior Instruments
GmbH, Goettingen, Germany, 5 Lokey Center for Life Science and Engineering, Technion, Haifa, Israel,
6 Faculty of Biology, Technion, Haifa, Israel, 7 Department of Molecular Biosciences, Northwestern
University, Evanston, Illinois, United States of America, 8 Department of Anatomy and Cell Biology, Brody
School of Medicine, East Carolina University, Greenville, North Carolina, United States of America, 9 East
Carolina Diabetes and Obesity Institute, East Carolina University, Greenville, North Carolina, United States
of America, 10 Faculty of Computer Science, Technion, Haifa, Israel, 11 Institute of Dental Sciences,
Faculty of Dental Medicine, Hebrew University of Jerusalem, Jerusalem, Israel

* peli@hadassah.org.il

## Abstract

Germ cell differentiation in the male testis involves extensive phenotypic, transcriptional, and epigenetic modifications, which are essential for producing functional spermatozoa. Among all organs, the testis exhibits the highest baseline physiological levels of L-2-hydroxyglutarate (L-2HG), yet its role in male germ cell development remains unknown. Here, we reveal that L-2HG is synthesized during the pachytene and diplotene stages of meiosis by the testis-specific enzyme lactate dehydrogenase C (LDHC). Surprisingly, LDHC translocates into the nucleus, localizing along the synaptonemal complex and at centromeres. L-2HG, produced by LDHC, regulates centromere condensation and heterochromatin organization via multiple mechanisms, including chromocenter clustering, centromere and chromocenter condensation, and modulation of satellite RNA expression. These effects are rapid, specific to L-2HG, and independent of histone methylation changes. Acute depletion of L-2HG in vivo results in centromere dysfunction and activation of the spindle assembly checkpoint (SAC), suggesting the possible role of this metabolite in ensuring proper chromosome segregation.

**Data availability statement:** RNAseq deposits can be found at NCBI with accession numbers GSE162740, GSE169014, GSE169015 and GSE238241. Pride-proteomics can be found at ProteomeXchange, with accession numbers PXD039014 and PXD039015.

**Funding:** EP was supported by a grant from Dr. Miriam and Sheldon G Adelson Medical Research Foundation (AMRF). URL: http://www. adelsonfoundation.org/AFF/index.html CBG was supported by a grant from NIH/NICHD R01HD110170. URL: https://www.nichd.nih. gov/grants-contracts The funders had no role in study design, data collection and analysis, decision to publish, or preparation of the manuscript.

**Competing interests:** I have read the journal's policy and the authors of this manuscript have the following competing interests: E.P. received compensation for speaking engagements and advisory roles, with payments directed to his affiliated university, from Roche, AstraZeneca, Novartis, and MSD. These activities were not related to the current manuscript.

## Author summary

2-Hydroxyglutarate (2HG) is recognized as an epigenetic regulator in cancer and immune cells. It modulates cellular metabolism by inhibiting alpha-ketoglutarate–dependent enzymes, which play crucial roles in processes such as DNA and histone demethylation. 2HG exists in two enantiomeric forms, D-2HG and L-2HG, each with distinct biological effects. D-2HG is strongly associated with IDH mutations found in certain tumors, notably gliomas and acute myeloid leukemia. In contrast, L-2HG levels increase in response to hypoxia, a common feature across diverse cell types. This elevation supports cellular adaptation to hypoxia by modulating energy metabolism and maintaining redox balance. Here, we report a novel mechanism of L-2HG function in mouse spermatogenesis, distinct from its previously described roles. The mouse testis contains 10- to 20-fold higher levels of L-2HG compared to most other tissues. Our study reveals that this high concentration of L-2HG is confined to specific stages of spermatogenesis, coinciding with the high expression of testis-specific lactate dehydrogenase C (LDHC), the enzyme responsible for L-2HG synthesis. By modulating L-2HG levels in cells at these stages, we demonstrate that L-2HG is critical for establishing the proper conformational organization of centromeres and the heterochromatin in which centromeres are embedded. Importantly, the effects of L-2HG that we observed are rapid and independent of its known epigenetic functions.

## Introduction

Germ cell differentiation in the testis involves profound phenotypic, regulatory, metabolic, and epigenetic changes. These changes are orchestrated to ensure the faithful transmission of genetic information to the next generation through a complex, multistage process that begins with diploid spermatogonial stem cells (SSCs) and culminates in the production of highly specialized haploid spermatozoa. Despite significant advances, relatively few studies have directly investigated germ cell–autonomous metabolic changes during mammalian male germ cell differentiation and meiosis [1,2]. A well-characterized example is the role of retinoic acid (RA) and its metabolism in spermatogonial differentiation prior to meiotic entry [3,4]. Since cellular metabolism can modulate the epigenetic landscape, influence gene expression, and even lead to heritable effects [5,6], it is plausible that metabolic regulation plays critical roles in both germ cell differentiation and the progression of meiosis.

In mammals, 2-hydroxyglutarate (2HG) is produced through the reduction of α-ketoglutarate (α-KG) in several physiological and pathological contexts [7]. It exists in two enantiomeric forms: D-2HG, which arises primarily from neomorphic activity of mutated isocitrate dehydrogenase (IDH1/2) [8], hydroxyacid-oxoacid transhydrogenase (HOT); [9], and the promiscuous activity of D-3-phosphoglycerate dehydrogenase (PHGDH); [10]. L-2HG, is generated by several enzymes, including lactate dehydrogenase A (LDHA) [11], the testis-specific isoform LDHC [12,13], and malate

dehydrogenase (MDH) [14]. These metabolites are normally degraded by enantiomer-specific 2HG dehydrogenases, and mutations in these enzymes lead to 2HG accumulation and associated metabolic disorders [15].

The pathological accumulation of 2HG in various tumors has led to its classification as an oncometabolite. Gain-of-function mutations in IDH1/2 result in high D-2HG levels and are implicated in gliomas, leukemias, cholangiocarcinomas, and chondrosarcomas [16,17]. Similarly, in clear cell renal cell carcinoma, copy number loss of L-2HGDH and the promiscuous activity of MDH cause L-2HG accumulation [18]. Both 2HG enantiomers are potent inhibitors of α-KG/Fe(II)-dependent dioxygenases, including the JmjC-domain-containing histone demethylases and the TET family of DNA demethylases [19,20], thereby linking 2HG accumulation to epigenetic dysregulation and tumorigenesis.

Beyond disease contexts, increasing evidence suggests that 2HG plays diverse and conserved roles across taxa. D-2HG is produced in bacteria [21] and yeast [22], while Drosophila and other dipterans synthesize large amounts of L-2HG during larval development [23,24]. In mammals, L-2HG contributes to T cell fate decisions during activation of $CD8^+$ T lymphocytes by promoting DNA and histone methylation changes that alter gene expression [25]. L-2HG also supports adaptation to hypoxia [11,26]: under low oxygen conditions, intracellular acidification broadens the substrate specificity of LDHA and MDH, enabling them to use α-KG to produce L-2HG. This process helps relieve reductive stress by consuming NADH.

Uniquely, the testis accumulates high levels of 2HG under physiological conditions, with L-2HG concentrations 10- to 20-fold higher than in most other tissues [12,27]. However, the specific stages of spermatogenesis at which 2HG accumulates, as well as its potential regulatory functions, remain poorly understood. In this study, we aimed to map L-2HG levels across distinct stages of male germ cell development and to uncover its functional significance. We found that L-2HG is generated at defined stages of meiosis and plays a critical role in shaping the structure and function of centromeres and heterochromatin, particularly at chromocenters. These findings may suggest a direct link between L-2HG and normal meiotic progression.

## Results

### Stra8-Tom mice enable isolation of viable germ cell populations from adult testes

To investigate the localization and function of L-2HG during spermatogenesis, we first developed a tool for isolating germ cells at distinct developmental stages. We fluorescently labeled the male germ cell lineage by crossing *Stra8-iCre* mice—which express Cre recombinase as early as the undifferentiated spermatogonia stage [28]—with *CAG-lox-stop-lox-tdTomato* reporter mice, generating "Stra8-Tom" mice. As expected, tdTomato expression was confined to the seminiferous tubules and did not overlap with the Sertoli cell marker SOX9, confirming specific labeling of the germ cell lineage (S1A Fig). Interestingly, tdTomato expression decreased markedly as germ cells progressed through development (S1B Fig).

Unexpectedly, tdTomato expression significantly decreased as germ cells proceeded through development (S1B Fig). This phenomenon likely occurred due to decreasing activity of the CAG promoter along the different stages, as ROSA-driven fluorescence (via crossing of ROSA-lox-stop-lox-YFP with Stra8-cre mice) was uniformly distributed (S1C Fig). We noted this gradual decrease could provide a means for isolating large and enriched populations of specific germ cell types at distinct stages of their development. FACS analyses of single cell suspensions prepared from testes of Stra8-Tom mice after dead cell removal (Forward scatter A vs. red fluorescence) revealed six discrete cell populations (Fig 1A) based on the stepwise decrease of tdTomato levels along with cell size differences.

To assess whether each population represented specific stages of the germ cell lineage and their cell purity, we analyzed DNA content with either Hoechst or propidium iodide (S1D Fig) coupled with immunostaining for known markers of different stages of the male germ cell lineage including DMRT1, ZBTB16/PLZF, SYCP3/SCP3, and γH2AX (S1E-H Fig). Immunostaining for cKit further separated undifferentiated from differentiating spermatogonia (Fig 1B). Our analyses revealed that each of the 7 cell populations was highly enriched for a specific stage of the germ cell lineage. Thus, we

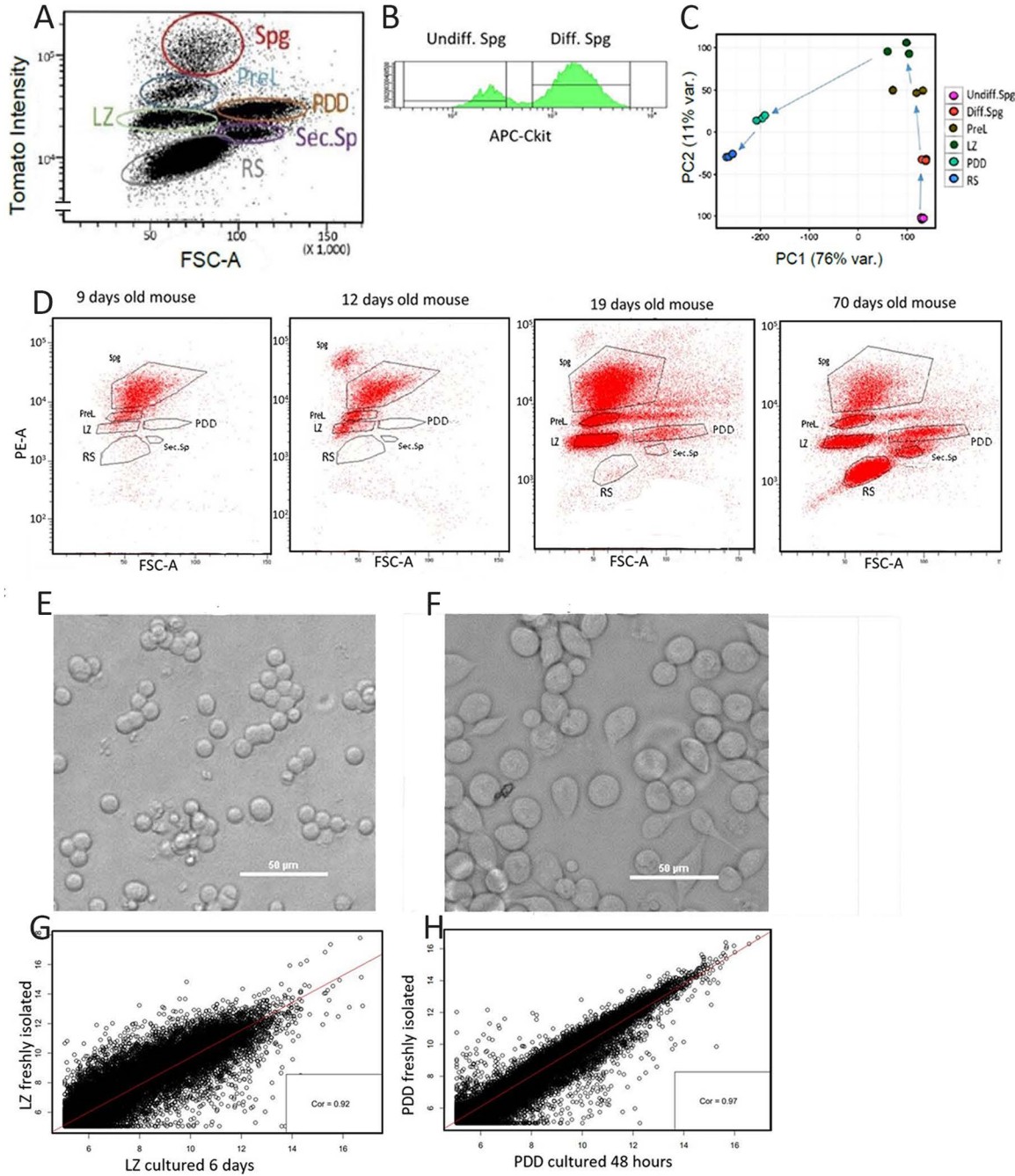

**Fig 1. Isolating highly purified populations of germ cells with Stra8-Tom mice.** (A) Representative FACS analysis of a Stra8-Tom testis single cell suspension. X-forward scatter; Y-red fluorescence. (B) FACS analysis of Spg cells stained with APC conjugated Ckit antibody. (C) PCA plot of RNA-seq data from indicated populations (n=3 for each population). (D) FACS analyses of testicular cell suspensions from the indicated ages revealed more dif-ferentiated cell populations gradually appeared with age. (E, F) LZ and PD cells were grown in culture for 6 days and 48 hours, respectively. Shown are representative phase contrast images. The notable change over time was that LZ cells tended to form clusters. Scale bar = 50 μm. (G) Gene expression (RNA-seq) from freshly isolated LZ cells vs. LZ cells cultured for 6 days. Pearson's product- moment correlation: r=0.92; p<2X10$^{-16}$. (H) Transcript levels (RNA-seq) from freshly isolated PDD cells vs. PDD cells cultured for 48 hours. Pearson's product-moment correlation: r=0.97; p<2X10$^{-1}$.

were able to sort undifferentiated (Undiff) and differentiating (Diff) spermatogonia (Spg), preleptotene (PreL), leptotene/ zygotene (LZ), pachytene/diplotene/diakinesis (PDD), secondary spermatocytes (Sec.Sp) and round spermatid (RS) populations. The composition and purity of each population is detailed in (S1 Fig). Next, we performed RNA-sequencing analyses (RNA-seq) on triplicate samples from the different cell populations (**GSE162740**). The Sec.Sp population was not analyzed due to cell number limitations. Unsupervised principal component analyses (PCA) revealed that each stage clustered separately and had ordered the cells along the spermatogenesis program (Fig 1C). The largest change in transcript abundance (PC1) was observed between LZ and PDD stages, both during prophase of meiosis I. Nearly 10,000 genes showed >2-fold change in transcript abundance between these two stages (S1I Fig). To further validate our method for separating germ cell populations, we harvested testis cells from Stra8-Tom mice at the indicated ages and subjected them to FACS analyses. The testes of 9-day-old male mice contained only spermatogonia; later stages gradually appeared in a semi-synchronous temporal pattern until the first testicular sperm was produced. Thus, during the first round of spermatogenesis in the developing mouse testis, more advanced cell germ cell populations gradually appeared with age, as expected (Fig 1D).

Next, we optimized culture conditions to maintain LZ cells for up to 6 days (Fig 1E) and PDD cells for up to 48 hours (Fig 1F) with minimal cell death of 21%±3 and 7%±1, mean±SE, respectively. mRNA expression of the myoid cell markers COL1A1 and ACTA2 showed negligible changes and remained low in LZ and PDD cells after 6 days and 48 hours in culture, respectively, when compared to freshly isolated cells. (S1 Table).

We subjected RNAs isolated from freshly sorted or cultured LZ and PDD cells to bulk RNA-seq analyses (**GSE169014**). Comparisons of the data revealed the cell populations maintained similar transcriptomes throughout the culture period ($r = 0.92$ and 0.97 for LZ and PDD populations, respectively,(Fig 1G and 1H). Thus, in addition to confirming the utility of the cultured cells to probe germ-lineage cell biology, this suggests the transcriptomes of these stages was relatively stable in these *in vitro* conditions, indicating the requirement for external signals for meiosis progression. This approach proved valuable for defining the localization and function of L-2HG, as detailed in the following sections.

## LDHC synthesizes L-2HG at specific stages of spermatogenesis

Previous studies reported that whole testis extracts harbored high content of L-2HG, yet the specific germ cell types that generated this metabolite were undefined [12,27]. Since Stra8-Tom mice provide us an opportunity to isolate sufficient numbers of viable cells for biochemical analyses, we sought to define the abundance of L-2HG at different stages of germ cell development. Thus, we analyzed extracts of metabolites from 4 major cell populations using LC- MS. This revealed the content of L-2HG increases 12- to 17-fold in PDD and RS populations as compared to LZ and Spg cells (Fig 2A). Chiral derivatization combined with LC-MS confirmed L-2HG levels were higher in PDD and RS cells vs. earlier stages, and showed that L-2HG was indeed the predominant enantiomer while D-2HG content was negligible (S2A Fig).

To assess the carbon source for L-2HG, we incubated LZ and PDD cells for 2 hours with 13C- lactate, 13C-glucose and 13C-glutamine and measured incorporation of each tracer into 2HG. As expected, the pool of L-2HG was ~25-fold higher in PDD compared to LZ cells. Two hours after incubating cells with labeled lactate, 20% of the L-2HG pool in PDD cells was labeled and appeared as the M + 2 isotopolog, whereas only negligible amounts of labeling were detected in LZ cells. Incubation with 13C- glucose labelled ~14% of the pool, while there was almost no incorporation of 13C carbons from 13C-glutamine. These findings support the following conclusions: (1) L-2HG is continuously expressed in PDD cells; (2) LZ cells do not actively generate L-2HG; and (3) the majority of L-2HG is derived from lactate, which is converted to pyruvate, enters the mitochondria, and feeds into the TCA cycle to produce α-ketoglutarate (αKG). αKG is then exported to extramitochondrial compartments most probably via the αKG/malate carrier (SLC25A11) [29], where it is subsequently converted to L-2HG (Fig 2B).

L-2HG is produced by several enzymes, including the three isoforms of lactate dehydrogenase (LDHA, LDHB, LDHC) and the two isoforms of malate dehydrogenase (MDH1, MDH2). Inspection of RNA-seq data revealed transcripts

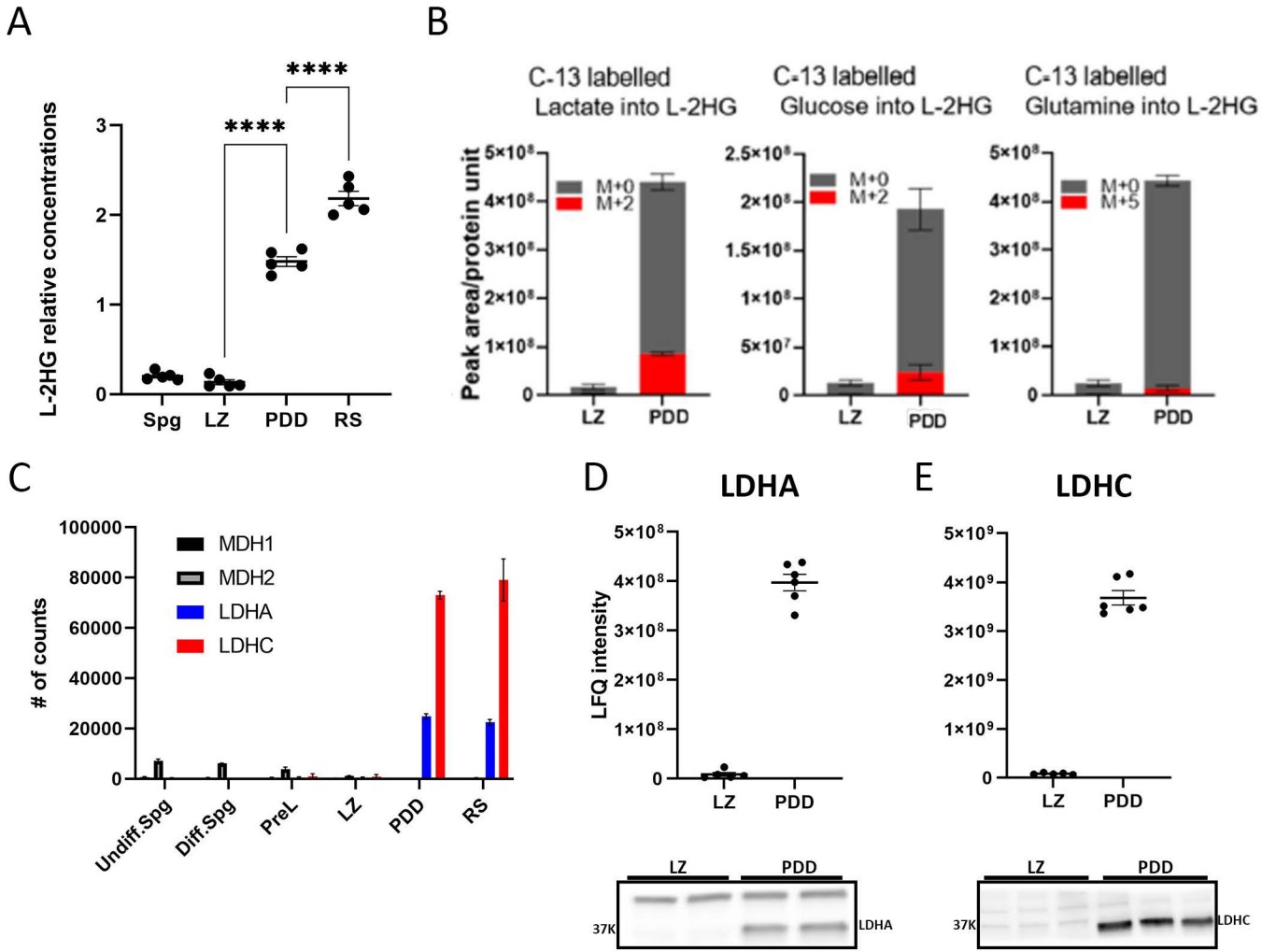

**Fig 2. 2HG was generated by LDHC in both PDD and RS cells.** (A) 2HG content (in the indicated populations) was measured by LC-MS in n = 5 independent experiments. Results were corrected for $10^6$ cells and cell volume. Cell volume measurements are provided in S2 Table. Mean±SE; One-way Anova-Tukey's multiple comparisons. (B) Incorporation of 13C-lactate-3, 13C-glucose-6, and 13C-glutamine-5 into 2HG isotopologs was measured using LC-MS. Peak area was corrected to protein content after 2 hours of incubation of LZ and PDD cells. Mean±SE of n = 4, 5, and 3 independent experiments of the tracing of lactate, glucose, and glutamine, respectively. (C) Expression values (counts) of LDHA, LDHC, MDH1, and MDH2 were derived from RNA-seq data, mean±SE of n = 3 samples. (D, E) The relative content of LDHA (D) and LDHC (E) derived from MS data(top) from n = 5 and 6 independent samples of LZ and PDD cells, respectively, with results presented as mean±SE. Bottom: immunoblots probed with the respective antibodies. An equal amount of protein was loaded from LZ and PDD cells.

encoding LDHA and the testis-specific LDHC were both markedly higher in PDD compared to LZ cells, while those for LDHB were not detectable in either population. Transcripts for MDH1 and MDH2 were both present at low levels and decreased during meiosis progression (Fig 2C). In line with the mRNA levels, results from both mass spectrometry of protein extracts and immunoblot analyses revealed LDHA and LDHC were markedly overexpressed in PDD vs. LZ cells (Fig 2D and 2E), thus theoretically both isoenzymes of LDH could synthesize L-2HG. Previously, it was shown that whole testis extracts from LDHC null mice [12,30] do not synthesize L-2HG and contain very low levels of L-2HG. Taken together, we concluded that LDHC, that is expressed in the germ lineage beginning at the pachytene stage is the enzyme responsible for L- 2HG production in PDD and RS populations. It is important to note that, according to the study by the Rabinowitz

group [12], the capacity to catalyze L-2HG synthesis by LDHC is restricted to mice. Other examined mammals do not possess this capability.

## The subcellular distribution of LDHC undergoes alterations during the progression of spermatogenesis

Numerous studies reported that specific cytoplasmic and mitochondrial metabolic enzymes can be transported to the nucleus, where they may function to generate metabolites involved in genome and/or epigenome regulation [31–36]. Given the known role of 2HG as a chromatin modifier, we investigated the localization of LDHC by performing immunostaining of cells from leptotene to RS, including cells in both anaphase I and II (Fig 3A, 3B and 3C).

Profound changes in the intensity and subcellular localization of this enzyme were observed throughout prophase I and during both meiotic divisions. As expected, LDHC immunostaining was clearly detectable in PDD cells and RS, but notably weak in LZ cells.

Remarkably, the localization of LDHC in cellular compartments changed with the progression of spermatogenesis. While LDHC was detectable in the cytoplasm of PD cells, it was also detectable in nuclei. In PD cells, LDHC was localized along the chromosome axes, as evidenced by colocalization with SCP3 (Fig 3A), and was particularly intense at the centromere region. This pattern was confirmed using high-resolution stimulated emission depletion (STED) microscopy (Fig 3D). Similar staining patterns were observed using a different anti-LDHC antibody targeting a different LDHC epitope and when staining nuclear spreads (S2B and S2C Fig). LDHC immunostaining of centromeres was also prominent in the diakinesis stage (Fig 3A) and in anaphase I but not in anaphase II (Fig 3A and 3B). During the second meiotic division and in RSs, LDHC was only detectable in the cytoplasm (Fig 3B and 3C). LDHC KO mice [30] were used to confirm staining specificity (Fig 3E).

Importantly, LDHA staining was also observed in nuclei of PD cells, but unlike LDHC, it was not localized to chromosomes (Figs 3F and S2D). Thus, while both LDHC and LDHA could produce nuclear L-2HG, only LDHC could play a putative role as a structural component of centromeres. To verify the presence of LDHC on chromatin, we extracted chromatin-bound proteins from LZ and PDD cells and submitted them for mass spectrometry analyses. Results from these analyses confirmed the presence of LDHC in the chromatin-bound fraction in PDD but not LZ cells. LDHA was not detectable in extracts from both cell types (Fig 3G). Principal constituents of the chromosome axis, including meiotic chromosome-specific proteins (SYCP1, SYCP3, and Hormad1), as well as ubiquitous proteins generally associated with chromatin, such as cohesins (SMC1b and SMC3) and DNA methyltransferases (DNMT1 and DNMT3a), exhibited diverse presence levels in LZ, PDD, or both populations [37,38].

We next assessed whether chromatin-bound LDHC retains catalytic activity, using pyruvate and α-ketoglutarate (αKG) as substrates, as previously characterized for the cytoplasmic enzyme [12]. Chromatin-bound LDHC, isolated from one million PDD cells, converted $1.2 \pm 0.03$ nmol of αKG to L-2HG (in the presence of 5 mM αKG) and $4.6 \pm 0.4$ nmol of pyruvate to L-lactate (in the presence of 1 mM pyruvate) over a 12-hour incubation period (mean ± SD, n = 3 technical replicates; S2E and S2F Fig). The substantially lower reaction rate observed with αKG compared to pyruvate is consistent with previous findings [12].

In conclusion, our findings reveal that L-2HG was generated significantly from pachytene to RS cell stages, largely by LDHC, and suggest that L-2HG was produced locally along the chromosomes in PDD cells, particularly in centromeric regions. The putative high local concentrations of L-2HG in centromeric regions in PDD cells led us to hypothesize that it may play a role in controlling centromere and heterochromatin organization and function.

## L-2HG guarded the spatial organization of centromeres and chromocenters of diplotene cells

Centromeres are highly dynamic structures that undergo unique changes in morphology to facilitate proper function in cell division [39,40]. To investigate the impact of L-2HG on centromeres, we aimed to acutely modulate its levels using oxamate, a well-known pan-LDH inhibitor [41]. Incubating PDD cells with oxamate for 48 hours resulted in a 10-fold decrease in L-2HG content (Fig 4A).

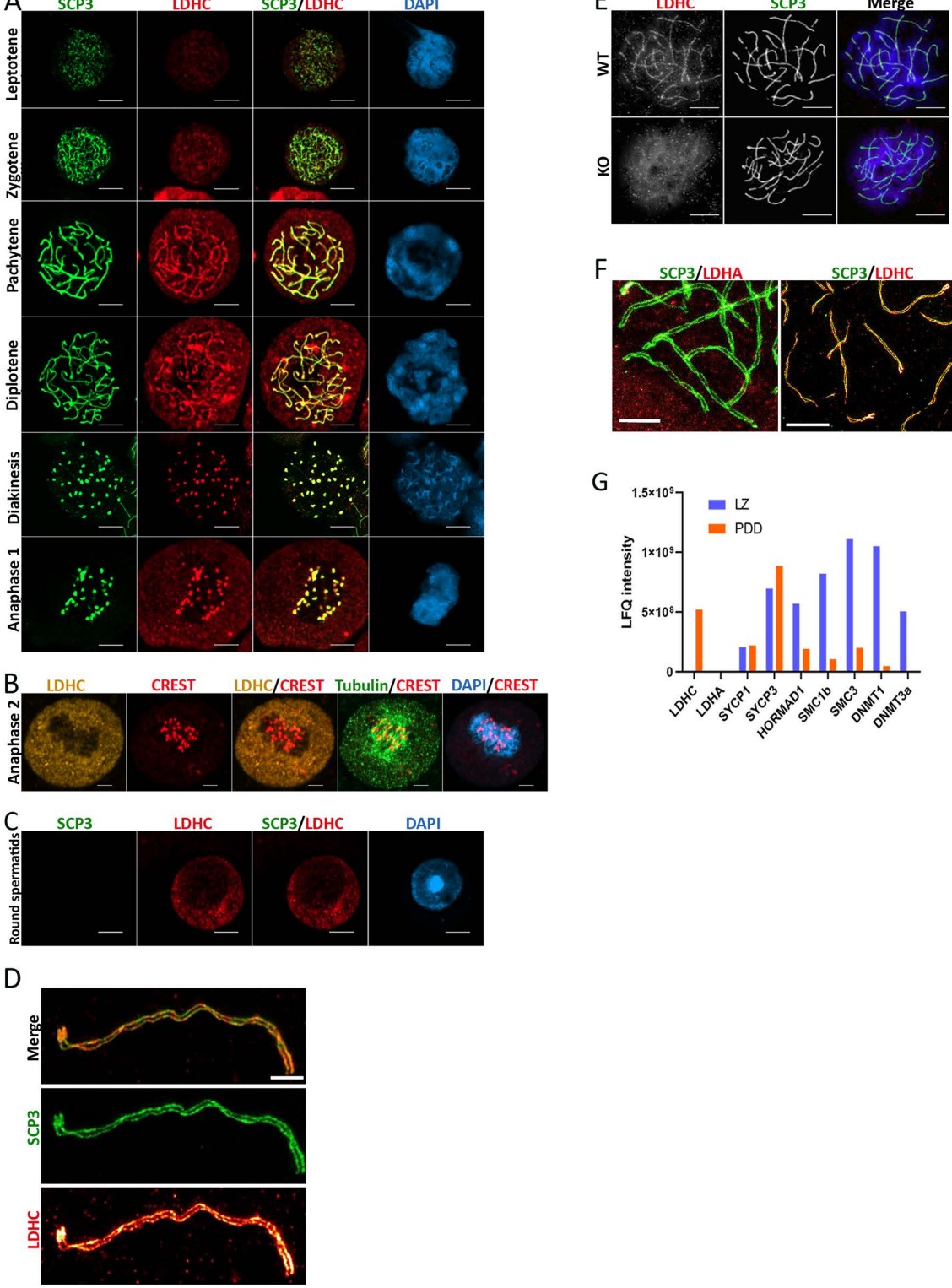

**Fig 3. LDHC localized along chromosomes and in centromeres.** (A-F) Immunostaining of freshly isolated testis cells for LDHC (199891-A antibody). (A-C) Confocal microscope images of germ cells at different stages of spermatogenesis, scale bar = 10 μm.(D) STED image of a pachytene chromosome, stained for LDHC and SCP3, scale bar = 1 μm. (E) Confocal microscope images of pachytene cells isolated from WT and LDHC-KO mice, scale bar = 10 μm. (F) STED image of pachytene chromosome stained for LDHA (left, scale bar = 2 μm) and LDHC (right, scale bar = 5 μm) with SCP3, (G)

Chromatin-bound proteins extracted from LZ and PDD cells were identified using MS. Note high content of LDHC in PDD population as compared to some proteins established as bound to meiotic/mitotic chromosomes in literature [37,38]. LDHA was undetectable. Results from one of n = 2 experiments are shown. All cells were isolated from Stra8-Tom mice except for results presented in (E).

To assess the effects of 2HG on centromeres of diplotene cells, we incubated cells for 24–48 hours with oxamate (to decrease L-2HG content) or with oxamate together with cell permeable L-2HG (octyl-L-2HG). Immunostaining with CREST antibodies revealed that oxamate treatment caused a 1.8-fold expansion of centromere cross-sectional area after 48 hours (Fig 4B and 4C). These measurements were performed for diplotene cells, as centromeres were more developed, and CREST immunostaining was more prominent at this stage compared to pachytene cells (S3A Fig). Oxamate-induced centromere decondensation was prevented by adding octyl-L-2HG to the incubation medium, suggesting L-2HG impacts centromere condensation in diplotene cells.

Next, we acquired high-resolution images of these alterations using a SIM microscope and quantitated changes in the centromere area. The analysis was conducted in a blinded manner. This analysis confirmed the impact on centromere cross-sectional area and also revealed that reducing L-2HG levels resulted in a diffuse pattern of CREST staining. This contrasted with the condensed pattern observed in the presence of octyl-L-2HG in diplotene cells. (Fig 4D and 4E). Centromeres are situated within well-defined heterochromatinized regions in diplotene spermatocytes. These heterochromatin regions, recognized as chromocenters and distinguished by their DAPI-dense appearance (Fig 4F), are subjected to transcriptional silencing by the repressive histone mark H3K9me3 [42]. This histone mark is bound by the heterochromatin protein HP1α [43].

Chromocenters, particularly prominent in the diplotene stage (S4B Fig), provide an opportunity to investigate the effects of L-2HG on both chromocenters and the centromeres embedded within them. Additionally, this setup allows for the assessment of potential interactions between these two nuclear sub-compartments.

Diplotene cells incubated for 24 hours with oxamate showed enlargement of chromocenter area, which was observed with both HP1α (Fig 4G and 4I) and H3K9me3 (Fig 4H and 4K) immunostaining; increases were 1.9- and 1.6-fold, respectively. This phenotype was rescued by octyl-L-2HG and was enantiomer-specific. The enlargement of heterochromatin area upon L-2HG depletion could be mediated either by an expansion of heterochromatization to lateral nucleosomes or by decreased compaction of pre-existing HP1α/H3K9me3 marked nucleosomes. To resolve between these two options, we measured Hp1α immunostaining intensity in chromocenters. We detected a 1.3-fold decrease in HP1α immunostaining intensity under oxamate treatment (Fig 4J) however we did not observe a significant change in the total amount of HP1α in the different treatment groups using immunoblot analysis (Fig 4M and 4N) thus implying the decompaction of heterochromatin. Notably, it was previously reported that the decompaction of heterochromatin can provoke aberrant transcription from this transcriptionally muted region [44].

Similar to results from a study with UV-induced heterochromatin decompaction in mitotic mouse embryonic fibroblasts [45], we found that H3K9me3 intensity was not decreased under oxamate treatment (Fig 4L), suggesting activation of H3K9 silencing-based methylation. These findings were confirmed by an experiment in which we administered oxamate to mice and subsequently examined H3K9me3 heterochromatin immunostaining in diplotene cells (S3C-F Fig). In this experimental setup, we observed an increase in the H3K9me3-stained area in diplotene cells isolated from oxamate-treated mice, similar to cells exposed to oxamate in culture. The reduction in the immunostaining intensity of H3K9me3 was notable (1.2-fold, S3D Fig), although much smaller than the decrease in DAPI intensity within the same chromocenters (1.8-fold, S3E Fig). The ratio of H3K9me3 intensity to DAPI intensity showed a significant increase (1.5-fold, S3F Fig), indicating an enhancement in the methylation of histone H3-lysine 9.

In spite of the generally silenced transcription in heterochromatin, the satellite sequences in these regions are transcribed into satellite RNAs. Transcription of satellite RNAs is known to be regulated along the cell cycle, and satellite RNAs were shown to be important for centromere function [46,47]. We hypothesized that L-2HG could upregulate satellite RNA levels.

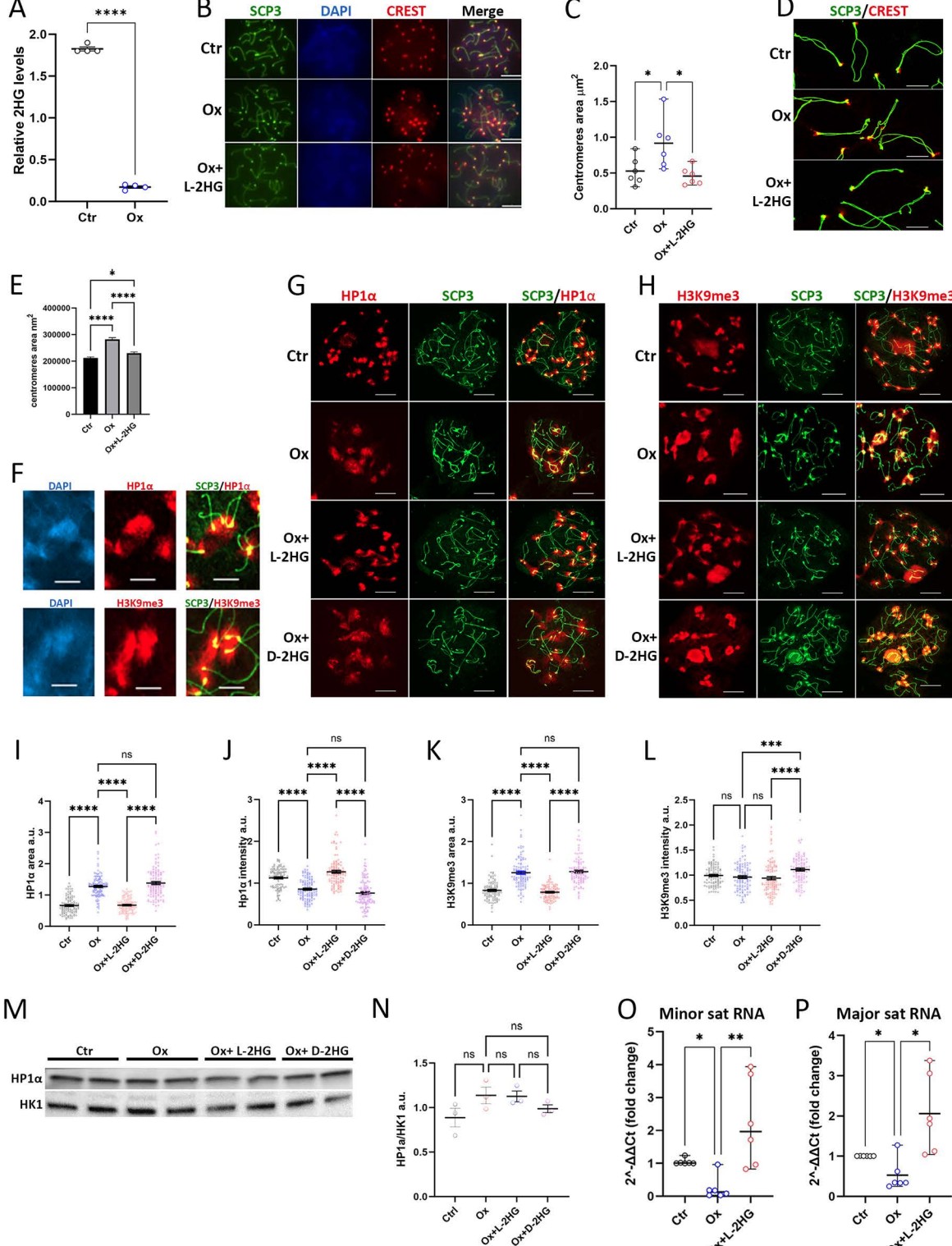

**Fig 4. L-2HG guarded spatial organization of centromeres and chromocenters in diplotene cells.** Isolated PDD cells were subjected to different incubation durations and treatments. Specifically, cells were incubated for 48 hours (A-E and O, P) or 24 hours (G-N) with either vehicle, 24 mM oxamate, or 24 mM oxamate with 0.3 mM cell-permeable octyl-L/D-2HG, as indicated. (A) Relative concentrations of 2HG were quantified using LC-MS.

Data represent mean±standard error (SE) from n=4 experiments, with significance determined by t-test. (B) Confocal images of representative diplotene cells were captured. (C) Centromeres areas from images (in B) were measured on CREST-immunostained nuclear spreads using NIS Elements Analyzer software. Each data point represents the mean of n=200-400 regions of interest from n=6 different experiments, with median and range displayed. Bonferroni correction was applied for multiple comparisons, one-tail. (D) Representative high-resolution Zeiss SIM3 images of nuclear spreads immunostained with antibodies against SCP3 and CREST. Scale bar=1 μm. (E) Quantification was performed to define CREST areas from images (in D). Mean±SE of ~n=600 centromeres in each group was determined using one-way ANOVA with Tukey's multiple comparisons test. (F) HP1α and H3K9me3 demarcate chromocenters, while diplotene chromosomes were immunostained for SCP3. Scale bar=5 μm. (G-H) Representative images were captured of chromocenters under different incubation conditions were immunostained with antibodies against HP1α or H3K9me3. (I-L) Quantitation of immunostained areas and the intensity of chromocenters of diplotene cells. Each data point represents the total area of one cell immunostained with HP1α antibody (I, J) or with H3K9me3 antibody (K, L). Mean±SE of ~n=100 cells in each group was determined using one-way ANOVA with Tukey's multiple comparisons test. (M-N) Immunoblot analyses for HP1α content in whole cell protein extracts of PDD cells normalized to HK1, representative of n=2 experiments. (O-P) qRT-PCR performed for minor (O) or major (P) satellite RNA. Data represent a median with a range from n=6 experiments, with Bonferroni correction applied for multiple comparisons, one-tail.

Indeed, levels of both minor and major transcripts significantly decreased, 10- and 3-fold respectively, in oxamate-treated PDD cells. Addition of octyl-L-2HG rescued this phenotype (Fig 4O and 4P). This effect was also recapitulated *in vivo* in germ cells from oxamate-injected mice (S3G and S3H Fig). In accordance, the above-mentioned study of UV- induced heterochromatin decompaction in mitotic mouse embryonic fibroblasts similarly reduced satellite RNA levels [45].

Changes in L-2HG neither influenced the area of diplotene nuclei (S4A Fig) nor several characteristics of chromosomes, including their length and width and the number of MLH1 foci (S4B-E Fig). LDHC KO mice had low testis levels of L-2HG since LDHC is the enzyme which catalyzes the conversion of α-KG into L-2HG [12]. Thus, we measured the area of diplotene cells' centromeres and heterochromatin in both wild type and LDHC KO mice. We did not observe significant changes in these parameters (S4F and S4G Fig), unlike in oxamate-injected mice (S4H and S3C Fig). Thus, mice with reduced L-2HG levels appeared capable of adapting to this situation, unlike the effect of acute depletion of L-2HG.

The simultaneous decompaction of centromeres and chromocenters observed under oxamate treatment suggests that L-2HG plays a role in regulating the morphology of these two structures through direct or indirect mechanisms. It is plausible that the influence on one entity can affect the other.

## L-2HG and LDHC guarded the morphology of centromeres in diakinesis

Next, we assessed the effect of oxamate treatment on centromeres in diakinesis cells, in which centromeres were fully developed to facilitate the first meiotic division. Based on the sequential assembly of centromeric proteins, it was only during the diakinesis stage that the appropriate arrangement of outer kinetochore proteins takes place [48].

While in general, it was difficult to obtain sufficient numbers of cells in diakinesis, FACS sorting of *Stra8*-TOM mice enabled us to collect sufficient numbers of diakinesis stage cells, which were assessed for immunostaining using antibodies against CREST and several CENPs (S3A Fig), as well as LDHC (Fig 3A). We studied the effect of L-2HG on the morphology of diakinesis centromeres (Fig 5A). Briefly, mice received a single intraperitoneal injection of either PBS or 1.3 g/kg oxamate and were sacrificed 24 hours later. Diakinesis cells from both groups of mice were isolated, and those from the oxamate-injected mice underwent a 10-minute incubation with either vehicle, octyl-L-2HG or octyl-D-2HG. Subsequently, the cells were processed for meiotic spread preparation and immunostaining. Oxamate treatment *in vivo* caused a significant enlargement of centromere area, measured using CREST antibodies, in diakinesis cells (Fig 5B).

Remarkably, even a 10-minute incubation of these cells with octyl-L-2HG at 32°C, just before chromosome-spread preparation and fixation with paraformaldehyde, was sufficient to revert centromere size and compaction by 1.8-fold (Ox vs. Ox+octyl-L-2HG). In contrast, a similar incubation with the octyl-D-2HG enantiomer had a very minor effect of 1.1-fold (Ox vs. Ox+octyl-D-2HG) (Fig 5C and 5D), indicating a stereospecific effect. Measurement of staining intensity under different treatments showed no consistent changes in CREST staining. In three of six experiments, oxamate treatment of mice increased CREST intensity; in another three experiments, no change occurred. The increase in CREST area was

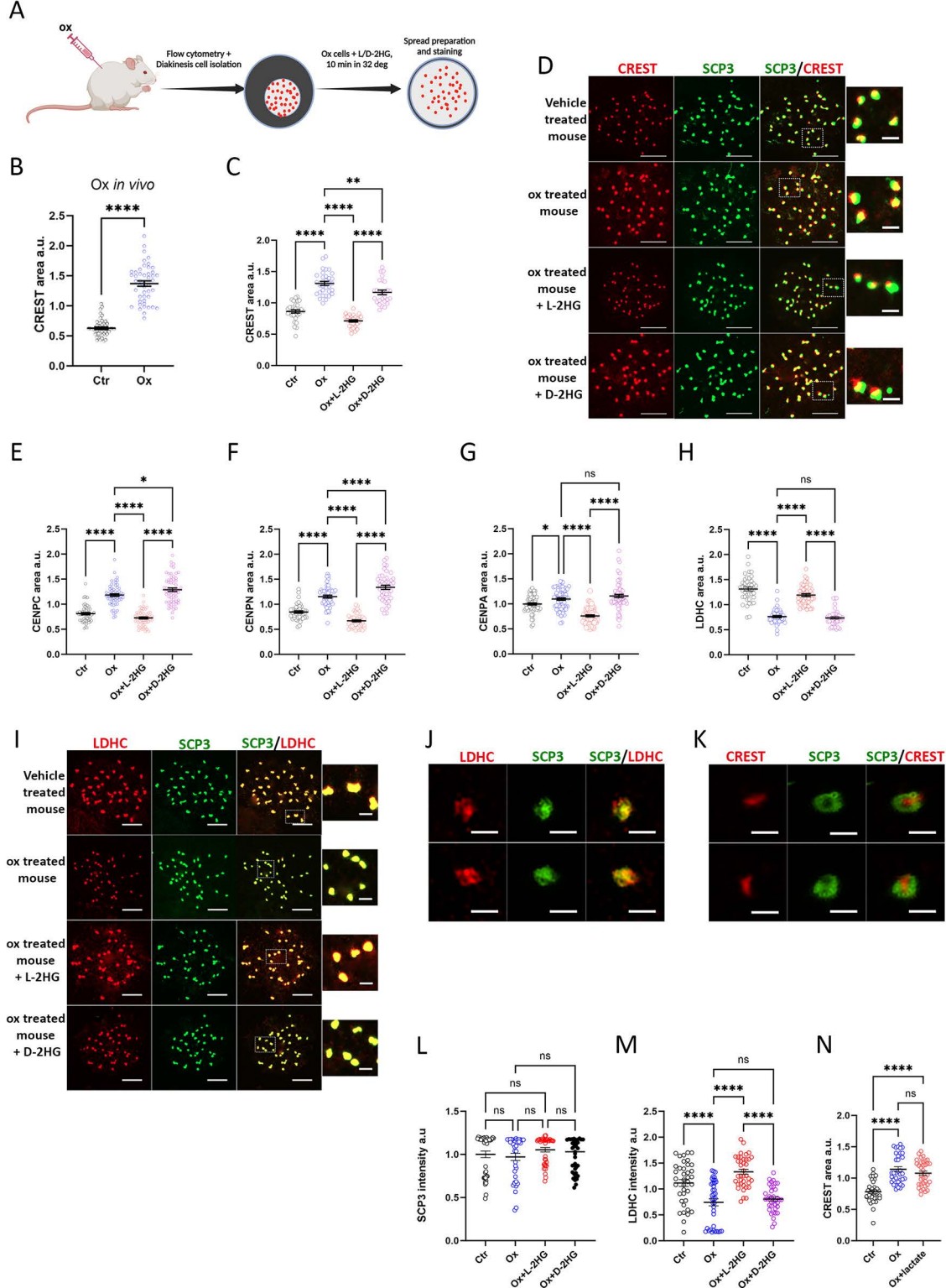

**Fig 5. L-2HG and LDHC guarded centromere morphology in diakinesis cells.** (A) Experimental Design (Created with Biorender). This experiment aimed to study the effect of 2HG on compacting oxamate-decompacted diakinesis centromeres. Diakinesis cells were isolated from two groups: mice injected once with oxamate (1.3 g/kg) and sacrificed after 24 hours (Ox) and mice injected with vehicle-only (Ctr). Diakinesis cells from oxamate-injected

mice, displaying enlarged centromeres (see B), were subjected to a 10-minute incubation at 32°C with 0.6 mM octyl L/D-2HG, while cells from control mice were treated under identical conditions with vehicle only. After incubation, nuclear spreads were prepared and immunostained for various markers: CREST/SCP3 (B-D, K), LDHC/SCP3 (H-J), CENP-C (E), CENP-N (F), and CENP-A (G). Representative images of E-G are presented in (S5 Fig). Cross-sectional area measurements of centromeres immunostained with the above antibodies were conducted (B-C, E-H). Another group of diakinesis cells underwent the same procedure but were incubated with 5 mM lactate instead of 2HG, and centromeres were immunostained with CREST antibodies (N). Each point in (B-C, E-H, and N) represents the mean area of all centromeres in one cell, while in (L-M), each point represents the mean staining intensity of centromeres in one cell. Data are presented as mean ± SE of n = 30-60 cells in each treatment. Statistical analyses were conducted using t-test for (B) and Tukey's multiple comparisons test for (C-H and L-N). Immunostained areas were measured using Zeiss ZEN 3.3 software and expressed as a fraction of the mean of all experimental groups in arbitrary units (a.u). (D and I) Confocal images of centromere areas immunostained with CREST antibodies (quantified in C) and LDHC (quantified in H) are depicted. Scale bars in D and I = 10 μm. (J-K) High-resolution images of centromeres captured by the N-SIM microscope are provided, scale bar = 1 μm.

consistent in all 6 experiments, thus excluding the possibility that the observed increase in area was due to the increase in intensity.

CREST antibodies primarily recognize CENP-A, CENP-B, and CENP-C, components of the core centromeric nucleosome complex (CCNC) [49]. To identify components undergoing decompaction, we immunostained with antibodies against CENP-C, CENP-N, and CENP-A. CENP-C contains intrinsically disordered regions (IDRs) [50], making it prone to liquid-liquid phase separation. CENP-N was recently shown to promote centromeric chromatin compaction in human cell lines [51]. Interestingly, both CENP-C and CENP-N staining showed an identical pattern of changes to those observed when using CREST antibodies (Figs 5E, 5F, S5A and S5B). This pattern was not observed with CENP-A immunostaining (Figs 5G and S5C). Oxamate treatment slightly enlarged the CENP-A area by ~1.1-fold compared to 1.5- and 1.4-fold for CENP-C and CENP-N, respectively. This was expected, as CENP-A forms the centromere-specific nucleosome at the base of the CCNC, to which CENP-C, CENP-N, and other proteins bind [52]. The effect of oxamate was reversed by addition of exogenous octyl-L-2HG but not octyl-D-2HG. We observed a modest but statistically significant 1.2-fold increase in decompaction of the CENPN area induced by addition of octyl-D-2HG (Ox vs. Ox +octyl-D-2HG). Additionally, there was increased ability of octyl-L-2HG to compact the CENPA area, with a decrease of 1.3-fold below control values (Ctr vs. Ox + octyl-L-2HG). However, qualitative changes in all examined parameters were similar, indicating a stereospecific effect.

Surprisingly, oxamate treatment resulted in LDHC loss from the centromeres, an effect that was rapidly restored by adding octyl-L-2HG (Fig 5H and 5I). Thus, decompaction was associated with the deposition of LDHC from centromeres, and the repositioning of this enzyme depended upon L-2HG. It is worth noting that immunolocalization of LDHC was adjacent to SCP3, as we observed nearly complete overlap in the staining of these two proteins in high-resolution SIM microscopy (Fig 5J). This differed from CREST and SCP3 immunostaining, where separation was quite distinct (Fig 5K). Since LDHC and SCP3 were closely positioned, we investigated whether changes in LDHC localization influenced SCP3. There was no effect on the intensity staining of SCP3 under different treatments (Fig 5L), unlike changes in the intensity staining of LDHC, which followed changes in cross-sectional areas (Fig 5M and 5C). This points to reversible attachment-reattachment of the enzyme to centromeres (SCP3 or other proteins).

In the context of this study, it was intriguing to examine whether lactate, the canonical product of LDHC activity, exerted a similar effect as L-2HG. We conducted the same experiment outlined earlier, substituting the 10-minute incubation with 0.6 mM octyl-L-2HG with the addition of 5 mM lactate (Fig 5N). However, we observed no discernible impact on the CREST immunostained area. Consequently, only the L-2HG product of LDHC demonstrated the capacity to influence centromere conformation.

## Chromocenters and centromeres in round spermatids were also affected by L-2HG manipulation

Round spermatids (RS) exhibited a notably elevated concentration of L-2HG, surpassing that of the PDD population (Fig 2A). While their cytoplasm prominently stains with LDHC, this staining is absent in their nuclei and centromeres (Fig 3C).

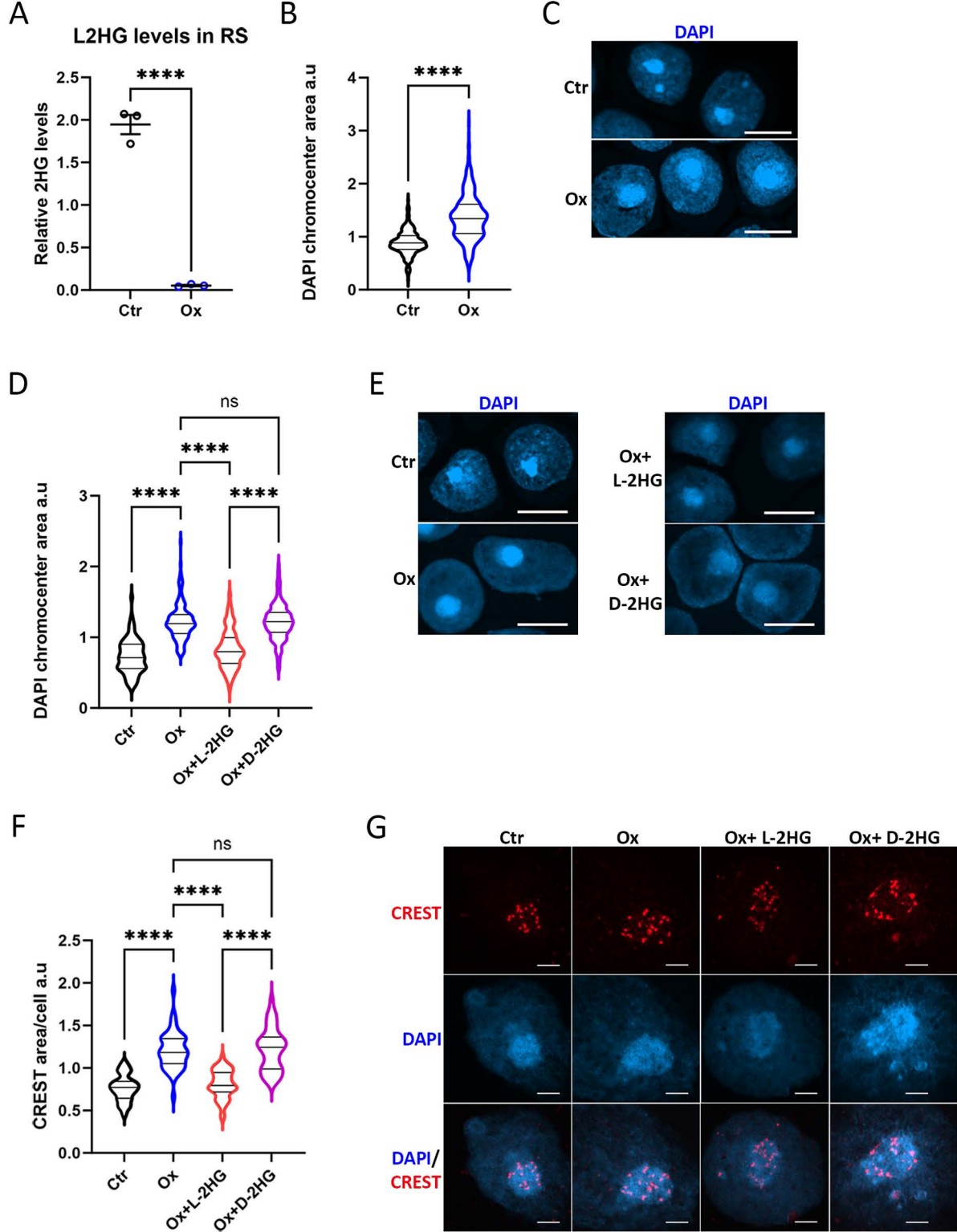

**Fig 6. Chromocenters and centromeres in round spermatids are also affected by L-2HG manipulation.** (A) RS cells were isolated using FACS-sorter and incubated for 24 hours with 24 mM oxamate. Relative concentrations of 2HG were quantified using LC-MS. (B-C) RS cells were isolated from mice injected once with oxamate (1.3 g/kg) and sacrificed after 24 hours (Ox) and from mice injected with vehicle (Ctr). (D-F) RS cells were incubated

with either vehicle, 24 mM oxamate, or 24 mM oxamate with 0.3 mM cell-permeable octyl-L/D-2HG, as indicated. (B-E) The DAPI-dense chromocenter area was quantified after cell adhesion to slides using cytospin and 10-second fixation with -20°C methanol. (F-G) For the evaluation of centromere area, RS nuclear spreads were prepared and stained with CREST. Statistical analyses were conducted using t-test for A (mean ± SE of n = 3 independent samples), Mann-Whitney for (B) (median with quartiles of n = 200 cells), and ordinary one-way ANOVA for (D and F) (median with quartiles; n = 150 cells and n = 40–50 cells in indicated treatments in D and F, respectively). (C, E, and G) Representative confocal images were taken and quantified in (B, D, and F), respectively. Scale bar = 10 μm.

This prompted an intriguing question about the potential impact of non-locally produced L-2HG on heterochromatin/centromere conformation. Our findings indeed support such an influence.

Analogous to the PDD population, incubating RS with oxamate for 24 hours significantly diminished L-2HG levels (Fig 6A).

A single injection of mice with oxamate results in a 1.5-fold increase in the cross-sectional area of RS chromocenters (one DAPI-dense chromocenter per cell) (Fig 6B and 6C). Furthermore, incubating RS cells with oxamate for 24 hours in culture leads to a similar 1.6-fold increase in the cross-sectional area of chromocenters (Fig 6D and 6E) and a 1.7-fold increase in total centromere area/cell (Fig 6F and 6G). Interestingly, the addition of octyl-L-2HG, but not octyl-D-2HG, prevents the expansion of heterochromatin and centromeres. These results indicate that nuclear production of L-2HG is not crucial for maintaining the conformation of heterochromatin/centromeres, at least in RS, as L-2HG synthesized by cytoplasmic LDHC can traverse the nuclear membrane and exert its effects.

### L-2HG manipulation induced global changes in the expression of genes controlling chromatin modifications

To investigate the potential impact of L-2HG on the transcriptome, we incubated PDD and RS cells with vehicle-only (DMSO 0.15%), oxamate 24 mM, and oxamate 24 mM supplemented with 0.3 mM octyl-L-2HG. Following the incubation periods (48 hours for PDD and 24 hours for RS), we isolated RNA from these cells and performed RNA-seq analyses (n = 3 in each group, **PDD: GSE169015 and RS: GSE238241**).

During our analyses, we specifically focused on genes fulfilling two conditions: 1) their expression changed when the levels of L-2HG were reduced (achieved through oxamate treatment); and 2) The addition of cell-permeable L-2HG to oxamate adjusted the expression towards their control levels. The aim was to isolate effects that could be specifically attributed to alterations in 2HG levels.

We employed a Likelihood Ratio Test (LRT) within the DESeq2 R package. To investigate relevant expression patterns in both the PDD and RS RNA-seq datasets, we focused on genes with significant (p < 0.05) transcript level changes. Subsequently, we applied K-means clustering to this subset of genes, generating clustering solutions from k = 2 up to k = 10. Through visual inspection, we determined that k = 6 (for RS) and k = 9 (for PDD) represented the optimal clustering solutions for our study.

In both the PDD and RS populations, we observed two distinct clusters of genes that exhibited the aforementioned changes in transcript abundance (Fig 7A and 7B).

In the PDD population, cluster 3, consisting of 310 genes, showed increased transcript abundance in response to oxamate treatment, which was subsequently balanced by the addition of L-2HG. Genes in cluster 9, comprising 291 transcripts, displayed the opposite effect: downregulated by oxamate and restored by adding L-2HG (Fig 7A and S1 Data).

In the RS population, we identified two clusters (clusters 1 and 2) with transcript changes from 1035 genes. Both of these clusters showed increased transcript levels upon oxamate treatment, but the addition of octyl-L-2HG counteracted this change, effectively restoring balanced transcript levels (Fig 7B and S2 Data).

We conducted enrichment analyses using the metadatabase GeneAnalytics [53], setting the pathway enrichment significance threshold at FDR < 0.05.

The enrichment analysis revealed a significant impact on transcript levels for genes associated with both chromatin organization and remodeling. Particularly, the analyses highlighted notable effects on transcript levels encoding proteins involved in histone modifications, including acetylation/deacetylation and methylation/demethylation (S3 Table).

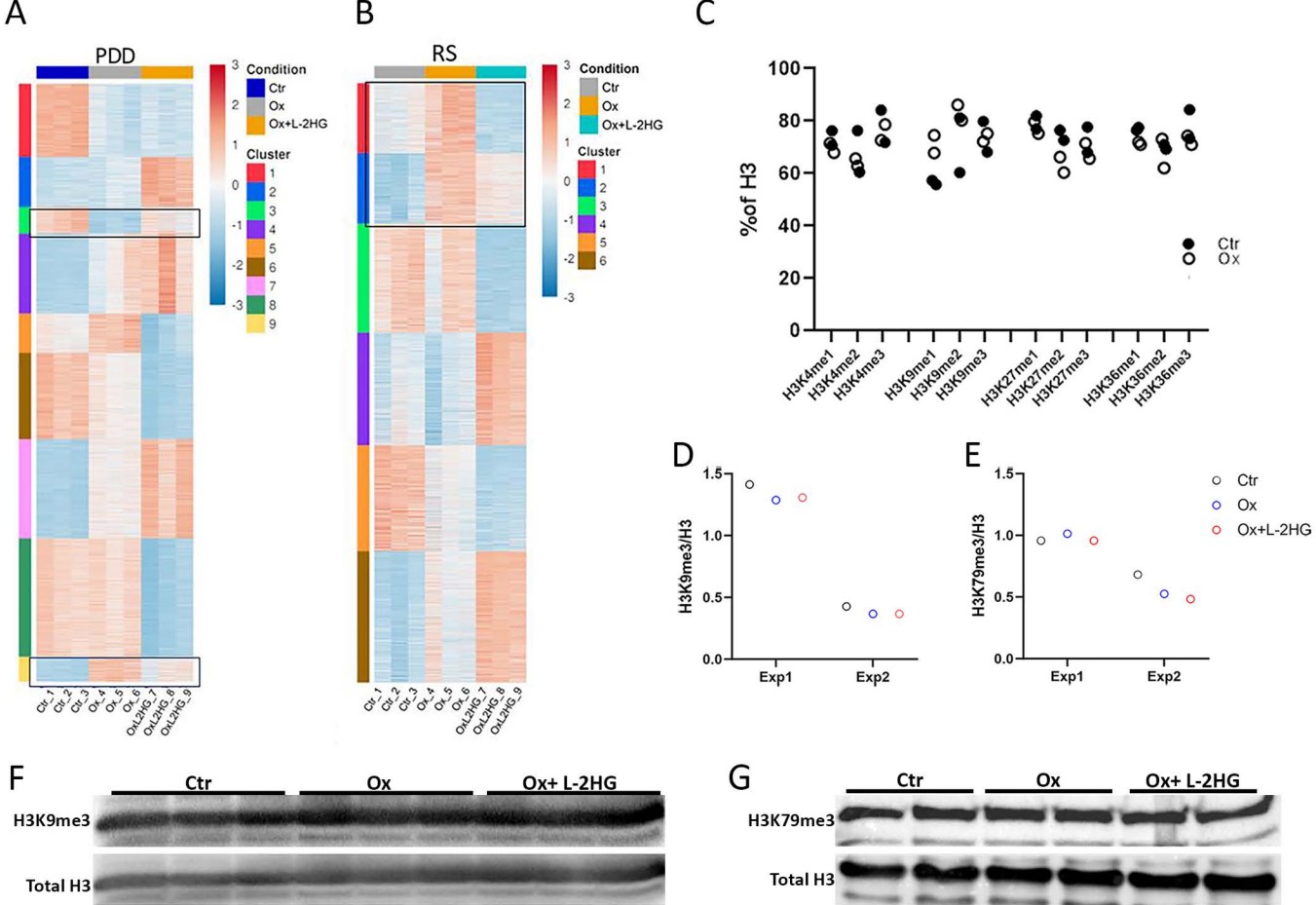

**Fig 7. L-2HG manipulation induced global changes in levels of transcripts encoding proteins controlling chromatin modification.** (A-B) PDD (A) and RS (B) cells were incubated in triplicates with vehicle, oxamate 24 mM, and oxamate 24 mM supplemented with 0.3 mM octyl-L2HG. Following the respective incubation periods (48 hours for PDD and 24 hours for RS), RNAs were extracted and subjected to RNA-seq analyses. Heat maps with optimal clustering were generated: n = 9 clusters in PDD and n = 6 clusters in RS. Clusters 3 and 9 in PDD cells, and clusters 1 and 2 in RS cells, contained transcript level changes for genes that can be attributed to alterations in L-2HG levels. See also S1 and S2 Data files and S3 Table. (C) Histone lysine modifications were measured using a histone extraction kit (ab113476) and multiplex colorimetric assay kit (ab185910) with 25 ng protein per well. Extracts were prepared from PDD cells incubated for 48 hours with either vehicle or 24 mM oxamate. D-G. PDD cells were incubated for 48 hours with vehicle, 24 mM oxamate, and 24 mM oxamate supplemented with 0.3 mM octyl-L-2HG. Histones were extracted and subjected to immunoblotting with antibodies against H3K9me3 (D, F) or H3K79me3 (E, G). (F and G) are representative images from one of n = 2 experiments. Quantitations of results from both experiments are presented in (D and E).

Interestingly, the analyses specifically highlighted effects on JmjC histone demethylases in both PDD and RS populations. It is noteworthy that the activity of these demethylases is known to be suppressed in the presence of 2HG however, a comprehensive examination of global levels of various histone modifications using an ELISA kit did not reveal consistent alterations in the overall levels of any modification in PDD histone extracts (Fig 7C). Histone H3 trimethylated lysine 79 and 9 are both situated in the chromocenters of the pachytene/diplotene cell population during spermatogenesis [54] We assessed levels of these two modifications after 24 hours of incubation with either the vehicle-only, oxamate, or oxamate with octyl-L-2HG using immunoblot analysis and found no significant changes (Fig 7D-G). It is plausible that the inhibition of JmjC histone demethylases induced intricate alterations in levels of transcripts encoding enzymes involved in histone modifications that resist changes at the global protein level.

**2HG is necessary for normal progression of meiosis**

Centromeres normally separate from each other in the diakinesis stage; on average, four centromeres remain paired in the diakinesis stage of male meiosis [55,56]. When examining diakinesis cells isolated from untreated mice, we noted an average of two unseparated pairs of centromeres, in-line with previous reports. Centromere pair numbers increased approximately 2-fold after oxamate treatment (Fig 8A and 8B).

This effect could also be observed in cultured diakinesis cells treated with oxamate for 24 hours and was prevented by the addition of octyl-L-2HG together with oxamate (S5D and S5E Fig). The above findings prompted us to examine the extent to which 2HG regulated centromere functions in PDD cells *in vivo*. As we have shown already, oxamate treatment *in vivo* recapitulated effects noted in the *in vitro* setting in PDD cells. Oxamate treatment increased the average centromere cross-sectional area of diplotene cells by 2.2-fold in treated mice (S4H Fig). Chromocenter area demarcated by H3K9me3 increased 1.7-fold (S3C Fig). Oxamate also reduced levels of minor and major satellite RNAs in PDD cells by ~5- and 2.5-fold, (S3G and S3H Fig), respectively. Notably, it was not possible to perform rescue experiments with cell-permeable octyl-L-2HG *in vivo*. Thus, all the effects of oxamate treatment on centromere and chromocenter morphology *in vitro* were recapitulated in the *in vivo* setting, allowing us to test whether L-2HG was necessary for the proper function of centromeres in live mice.

The meiotic prophase in male mice is extremely long, lasting about 2 weeks – compared with the 30–60 minutes duration of mitotic prophase; the pachytene-diplotene stages of mouse male meiosis last about one week [57,58]. Therefore, to assess the effects of L-2HG depletion on meiosis progression *in vivo*, we treated mice with oxamate for 7 days. Control mice were treated with vehicle-only and euthanized on day 8 of the experiment. Histological analyses of PAS-stained slides and immunostaining for cleaved caspase 3 did not reveal signs of toxicity (S6 Fig). Next, we immunostained testis sections with antibodies against histone H3 phospho-serine 10 (pHH3) – a marker of cells in late G2/M phases. Oxamate treatment resulted in a 2-fold increase in numbers of pHH3 positive cells (Fig 8C and 8D). This suggests either an increased proliferation rate or cell cycle arrest, as H3 remains phosphorylated in cells that are arrested in the G2M checkpoint. Cell cycle arrest due to centromere malfunction was executed by activation of the spindle assembly checkpoint (SAC) [59]. Thus, we co-immunostained testis sections with pHH3, MAD1 and CREST antibodies and quantified the number of pHH3 positive cells harboring MAD1 localized to the centromeres – an indication of SAC activation [60,61]. Remarkably, oxamate treatment increased the number of cells in which the SAC was activated by 6-fold (Fig 8E and 8F), supporting a role for L-2HG in ensuring the fidelity of centromere function in male meiosis. Taken together, our data suggest the high physiological levels of L-2HG present in PDD cells were necessary for proper assembly of the centromeric complex and its absence could result in centromere decondensation and malfunction during meiosis in the male germline.

## Discussion

The oncometabolite 2HG gathered attention when it was discovered to be highly abundant in tumors harboring activating mutations in IDH enzymes [8,16,17]. It is also upregulated in lymphocytes in specific states and in hypoxic cells where it governs cellular transcriptional responses [19,20,26]. While L-2HG is most abundant in the testis [12,27,62], its distribution in germ cells and whether it accumulates as a metabolic by-product or plays an active role in regulating germ cell biology remain unknown.

Here, we developed a flow cytometry-based method for isolating and culturing high numbers of viable germ cells of specific stages, similar to a previously published methods [63–65]; however, our method does not require retinoic acid (RA)-based spermatogenesis synchronization. Using this method, we report the expression of L-2HG-producing enzymes is restricted to specific stages of the first meiotic prophase and only initiates after completion of the zygotene stage. Accordingly, we detected high levels of L-2HG in PDD and RS stages, but not in earlier stages. We confirmed, using the pan-LDH inhibitor oxamate, that LDH is the major source of L-2HG in the testis. In accordance, it was previously shown that LDHC KO mice have low levels of L-2HG in the testis [12]. Remarkably, we now show that LDHC protein is markedly

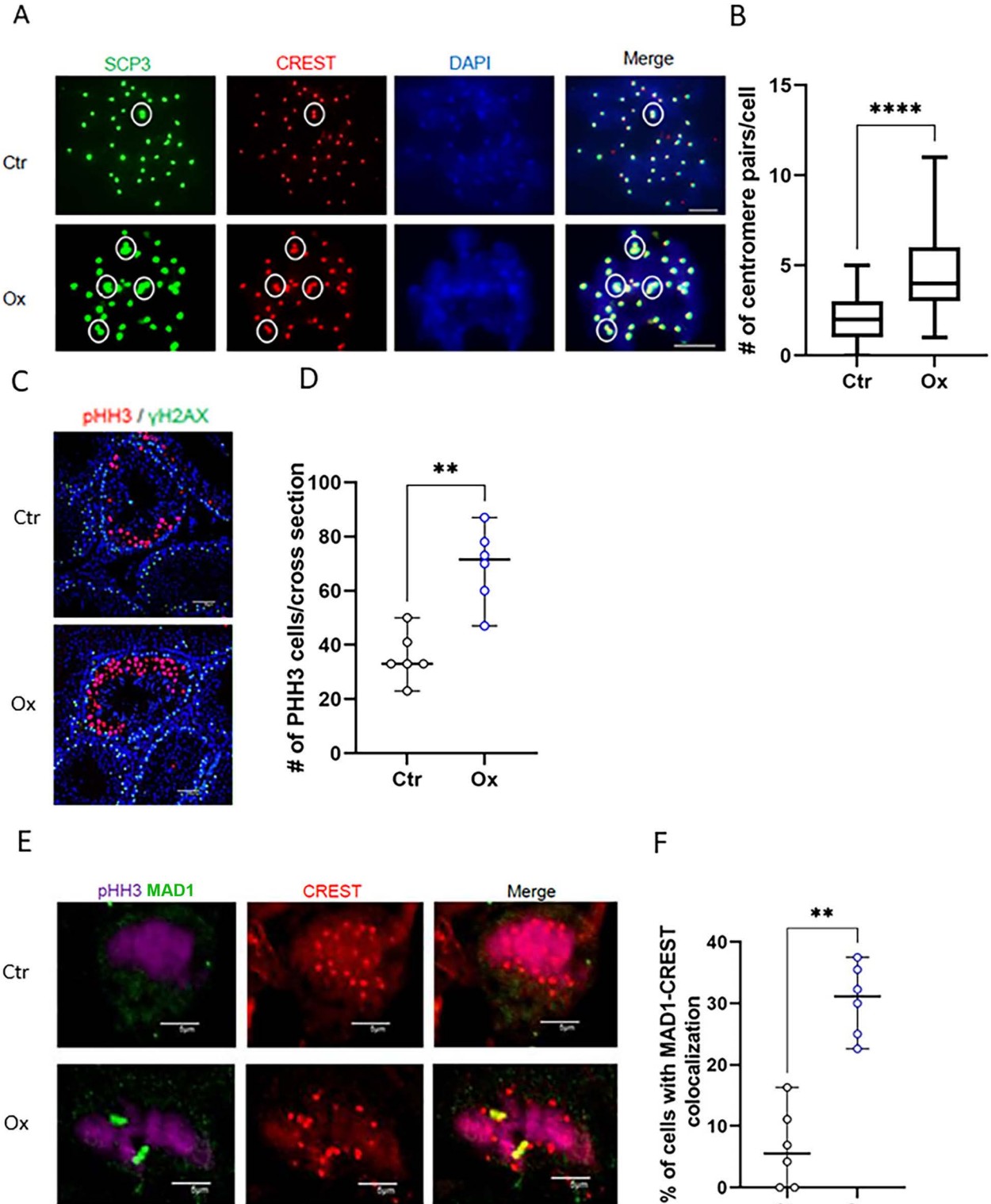

**Fig 8. Oxamate treatment caused meiotic centromere dysfunction and activation of the spindle assembly checkpoint *in vivo*.** (A-B) Mice were injected once with oxamate 1.3 g/kg or vehicle. Diakinesis cells present in the isolated PDD population were immunostained for antibodies against SCP3 and Crest. (A) Centromere pairs are marked with white circles, scale bar = 10 μm. (B) Number of centromere pairs were counted in n = 40 cells

in diakinesis, median with range, Mann-Whitney. (C-F) Mice were treated with daily oxamate injections (as above), or vehicle-only for 7 days (n = 6 experiments). (C) Paraffin sections were immunostained for pHH3, γH2AX and then counterstained with DAPI. Representative image, scale bar = 50 µm. (D) The numbers of pHH3-positive cells per cross-section were quantified. Each dot represents the mean of ~n = 15 cross-sections per mouse, with n = 6 mice per group. Median values with ranges were calculated, and statistical analyses were performed using the Mann-Whitney test. One experiment out of two is shown. (E-F) Paraffin sections were immunostained for PHH3 (purple), MAD1 (green) and Crest (red). pHH3 positive cells were assessed for the presence of overlapping staining for MAD1 and CREST. (E). Representative image, scale bar = 5 µm. (F) The percentage of MAD1-CREST overlapping cells was determined from pHH3-positive cells, with n ≥ 30 nuclei analyzed per mouse. Each data point represents a single mouse, with n = 6 mice per group. Median values with ranges were calculated, and statistical analyses were performed using the Mann-Whitney test.

increased at the PDD and RS stages and is associated with centromeres of PDD cells and along chromosomes in the PD stages. While we could not measure local concentrations of L-2HG in the centromere region, we speculate that the unique distribution of LDHC generates high concentrations of L-2HG in these subnuclear compartments. As LDHC KO mice have low levels of 2HG in the testis, it appears that LDHA is insufficient to produce 2HG in this tissue. Still, LDHA is highly expressed in PDD cells, but not in earlier stages. While LDHA is also detected in nuclei, it does not localize to centromeres or chromosomes.

Using immunostaining and proteomic analysis of chromatin-bound proteins, we found that LDHC—but not LDHA—is associated with chromatin. We also observed that chromatin-bound LDHC is enzymatically active, producing both L-lactate and L-2HG. It is plausible that locally synthesized L-lactate contributes to histone lactylation, as recently demonstrated in human and mouse somatic cells [66] and in mouse male germ cells during prophase [67]. Furthermore, the local production of L-2HG may influence satellite RNA expression, heterochromatin compaction, and histone methylation, as discussed below.

It is assumed that hypoxic conditions prevail within the seminiferous tubules, particularly closer to their lumen, since blood vessels are located in the intertubular compartment and do not penetrate the tubules themselves [68]. The elevated levels of L-2HG observed in PDD and RS germ cell populations could plausibly be attributed to the hypoxic microenvironment of the seminiferous tubules. Hypoxia has been shown to promote the LDHA-catalyzed conversion of α-ketoglutarate (α-KG) into L-2HG, a process that may stabilize HIF-1α [69].

However, this otherwise compelling hypothesis conflicts with existing data showing that in the testis, the majority of L-2HG is produced by lactate dehydrogenase C (LDHC), whose enzymatic activity is not regulated by acidic pH [12]. Furthermore, our transcriptomic data reveal the absence of Hif1a mRNA expression in the PDD population, a finding that is corroborated by immunostaining for HIF-1α expression [70]

Taken together, these findings suggest that the high levels of L-2HG observed in PDD and RS cells are primarily driven by the elevated expression of LDHC at both the mRNA and protein levels. Whether this upregulation is associated with hypoxia remains an open question.

We found that the levels of minor and major satellite RNAs expressed from pericentromeric regions depends on L-2HG. Satellite RNAs allow for proper centromere function during both mitosis and meiosis, presumably through maintaining the compaction of the centromere either via tethering or recruiting centromeric proteins or through induction of phase separation processes, similar to other long noncoding RNAs [39,46,47,71]. Both forced overexpression and under-expression of satellite RNAs cause defects in centromere morphology and chromosomes segregation in mitotic cells [72,73]. It was recently shown that satellite RNAs play a role in regulating the fidelity of meiosis [47,74], and their depletion activated the Spindle Assembly Checkpoint (SAC) in female meiosis [74].

The way by which L-2HG controls levels of satellite RNAs is not yet clear. This could be via classic transcriptional regulation of the satellite RNA genes, presumably by controlling local DNA or histone methylation. It is worth noting that we did not detect global alterations in histone methylation. Our mRNA-seq analyses of both PDD and RS populations, when subjected to 2HG depletion through oxamate treatment, revealed changes in the levels of transcripts encoding several histone-modifying enzymes, including JmjC histone demethylases. Among JmjC demethylases, KDM2B, KDM4A, KDM5B, and KDM5C

underwent transcriptional changes in both PDD and RS cells when L-2HG levels were modulated by oxamate and the addition of octyl-L2HG to oxamate-treated cells (S3 Table). Among these demethylases, the activity of KDM4A has been shown to be inhibited by L-2HG (IC50 = 26 μM) [19]. This enzyme demethylates H3K9me3 and H3K36me3. The former is known as a transcriptional suppressor [42], while the latter is known to function as both a suppressor and an activator of transcription [75].

Our data show that oxamate treatment increases expression of this gene in both PDD and RS cells, while the addition of octyl-L-2HG decreases it (S1 and S2 Data). Interestingly, oxamate-mediated L-2HG reduction likely de-represses enzymatic activity and upregulates mRNA expression.. However, we do not observe a decrease in H3K9 methylation at the global level under oxamate treatment; on the contrary, we observe a relative increase in H3K9 methylation in chromocenters (Figs 4L and S3F).

These results suggest that modulation of L-2HG levels triggers multiple and complex changes in histone-modifying enzymes, the impact of which cannot be simply predicted based solely on changes in gene expression.

It is also plausible that the changes in the total levels of satellite RNAs reflect downstream effects of heterochromatin condensation (as indicated by H3K9me3 and HP1α immunostaining), which could alter either their synthesis or decay rates. As indicated above, our results are similar to those described for de-compacted chromocenters of embryonal mouse fibroblasts where transcriptional silencing was prominent [45].

Many male mice with loss of various testis-specific genes exhibit normal fertility, likely due to functional redundancy [76,77]. Germ cells express 90.5% of all protein-coding genes compared to somatic cell types, which collectively express 59.9% of genes [78]. It is plausible to assume that this abundance of transcription in male germ cells favors compensation for the loss of activity of certain proteins. Therefore, we did not expect LDHC-knockout mice to phenocopy all the phenotypes in 2HG depleted spermatocytes, due to the activity of additional enzymes that metabolize L-2HG and function in the absence of LDHC. LDHC-deficient males produce sperm and lack meiotic defects thus it is evident that LDHC is not essential for meiotic progression.

Indeed, knockout mice showed no change in centromere and heterochromatin compaction. We maintain that the carefully conducted rescue experiments, showing that exogenous cell -permeable L-2HG corrects all the reported effects induced by oxamate, strongly argue for the specificity of our claims. Genetic knockout represents a chronic absence of LDHC, allowing for long-term adaptations. In contrast, oxamate inhibition is an acute effect (that we demonstrate in vivo and in culture after 24 hours already) that may not allow time for compensatory mechanisms to develop. Indeed, just 10 minutes of incubation with octyl-L-2HG are sufficient to reverse the decompaction effect of oxamate. In addition, oxamate is a general LDH inhibitor that affects multiple LDH isoforms, not just LDHC. Thus, the pronounced effect observed with oxamate may result from a broader inhibition of lactate dehydrogenase activity. It is also possible that an additional metabolite, structurally similar to L-2HG is generated in LDHC knockout mice which could act as a conformation altering metabolite on the same target protein/proteins as L-2HG. Further research is necessary to outline the degree of L-2HG loss in spermatocyte nuclei and delineate the general changes in the metabolic network in these cells in order to elucidate how LDHC-KO mice compensate for the likely lower levels of L-2HG in the nuclei of their spermatocytes.

The regulatory roles of L- and D-2HG in cancer and physiology are thought to be mediated through inhibition of α-KG-dependent demethylases [19,20]. Notably, the few studies that compared inhibitory activities of L- and D-2HG towards different histone demethylases did not observe marked differences between the two enantiomers [19,20]. Our observation that decompacted chromocenters undergo increased methylation of H3 lysine 9 despite the low level of L-2HG induced by oxamate treatment suggests that this metabolite did not control demethylation but rather compaction. It is well known that lactate produced by Sertoli cells is a major source of energy for some spermatogenic cell populations, suggesting that the oxamate effects are mediated by reduced energy production. Although the rescue of the oxamate effect by L-2HG argues against this hypothesis we cannot totally exclude the possibility that some of the effects are due to the energy imbalance.

The enantiomer-specific and rapid (=10 min) condensation of diakinesis centromeres by octyl-L-2HG at 32°C supports the possibility that L-2HG induced a conformational change in a specific target protein via binding to an allosteric

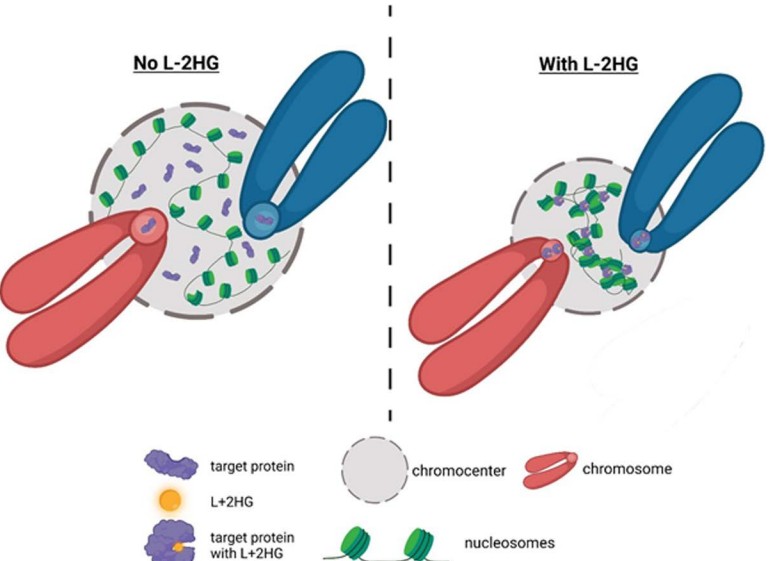

**Fig 9. Model for the effects of L-2HG on chromocenters and centromeres.** Upon binding of L-2HG, the target protein undergoes a conformational change. Some of the L-2HG-bound target protein translocates from the chromocenter to the centromere. The L-2HG-bound target induces condensation of nearby chromatin. This could occur either via protein-protein interactions or via induction of polymer-polymer phase separation as was shown previously for HP1α. This figure was created with BioRender.com.

regulatory site according to a lock-and-key mechanism. Interestingly, phosphorylation of a structured region of HP1α, a key effector of heterochromatinization, exposed a disordered region in this protein to form homotypic polymer-polymer phase separated (PPPS) chains and affect heterochromatin structure [79]. We hypothesize that a similar effect occurs upon binding of L-2HG to a specific target protein. Our data show LDHC localization to the centromere requires normal levels of L-2HG, thus raising the possibility that this protein could be the L-2HG binding protein. We hypothesize that the conformational change induced by binding to L-2HG renders the target protein more capable of forming high order structures resulting in condensation of the chromocenter and centromere (Fig 9).

The compacting effect of L-2HG on heterochromatin observed in mouse male germ cells in our study is consistent with findings reported in *Drosophila* larvae [23]. In the *Drosophila* study, it was shown that the *Drosophila* homolog of LDH (dLDH) synthesizes L-2HG and promotes heterochromatin formation. Mutations in this enzyme act as strong recessive suppressors of position-effect variegation (PEV) [80]. These findings suggest that, beyond its role as an oncometabolite, 2HG plays important roles in chromatin regulation across diverse organisms, as evidenced by studies in mice, *Drosophila* [23], and humans [25].

To our knowledge, the regulation of centromere compaction by a metabolite synthetized at this specific stage of germ cell development is quite unique, however it is possible that similar metabolic effectors will be found in other systems. Identifying target protein(s) whose conformation is regulated by L-2HG will reveal whether the mechanism we discovered in the male germline may be operative in other physiological or pathological conditions.

## Methods and materials

### Methods

**Ethics statement.** All mice (except LDHC-KO mice) were bred at the Hebrew University- Hadassah Medical School (Jerusalem, Israel), and protocols for their care and use were approved by the Institutional Animal Care and Use Committee of the Hebrew University (approval #MD-22-16877-2).

LDHC-KO mice were bred at East Carolina University (Greenville, NC, USA), and their use was approved by the East Carolina University Institutional Animal Care and Use Committee (approval #A3469-01).

**Mice.** Stra8-iCre (B6.FVB-Tg (Stra8-icre)1Reb/LguJ) male mice (stock 017490), tdTomato (B6;129S6-Gt(ROSA)26Sortm9(CAG-tdTomato)Hze/J females(stock 007905) and EYFP (B6.129X1- Gt(ROSA)26Sortm1(EYFP) Cos/J females (stock 006148) were purchased from the Jackson Laboratory. LDHC knockout mice, were developed by Erwin Goldberg and E. Mitch Eddy [30]. All mice are on C57Bl/6 background. Hemizygous Stra-iCre males were mated with either homozygous tdTomato or with homozygous EYFP females, and males which were positive for Stra-iCre were used as a source for testis tissue. Genotyping was done by quantitative PCR according to the suppliers' protocol. (https://www2.jax.org/protocolsdb/f?p=116:5:0::NO:5:P5_MASTER_PROTOCOL_ID,P5_JRS_CO DE:29550,017490). When indicated, mice were treated with 1.3 g/kg oxamate once daily, for the indicated periods of time.

## Cell isolation

Prepubertal and adult (>10 weeks old) mice were euthanized with pental followed by cervical dislocation. Testis were removed, put into cold Hank's solution and seminiferous tubules were released from the tunica albuginea. Tubules were examined with fluorescence microscopy to verify tdtomato expression. Tissue digestion was performed in two steps. First, tubules from two testes were put into 20 ml medium (DMEM/F12) with collagenase (1 mg/ml) and DNase (0.28 mg/ml), then gently dispersed by pipetting using sequentially 25, 10, and 5 ml pipettes. After incubation for 2 min in room temperature and washing with medium, 30 ml of trypsin diluted 2:1 with medium and containing 0.2 mg/ml DNase was added. Tubules were incubated at 32°C for 5 min with frequent gentle inversions of the tube. After increasing of DNase concentration to 0.93 mg/ml and additional 3 min incubation in room temperature, 2.8 ml FBS was added to stop trypsin activity. The resulting suspension was passed through 40 µm cell strainer, washed in medium and incubated with dead cells removal beads for 15 min in room temperature. Live cells were collected according to the manufacturer's instructions and washed in PBS with 5% serum followed by dispersal in sorting buffer. When separation between Undiff and Diff Spg was required, live cells were pre-incubated for 10 min on ice with 1% rat serum and 1.2% FC in PBS, to block non-specific binding, continued by incubation with APC-cKit antibody (1: 900) for 20 min on ice.

## Flow cytometry sorting

After live cell isolation, cells were washed in 2% dialyzed serum in PBS, resuspended in "sorting buffer" at a density of 6 million cells/ 1 ml. During sorting, cells were collected into 5 ml tubes, which were kept overnight in 4°C, inverted with 2ml of different "collecting buffers" to prevent cell loss. Cells were sorted using a BD FACSAria III sorter. Debris exclusion was done based on light scattering. tdTomato intensity (yellow-green laser, bandpass 582/15) of cells versus forward scatter (FSC) was used to differentiate between the different populations. Doublets exclusion in each population was performed on the basis of FSC width/area (FWA), followed by side scatter (SSC) width/area (SWA) and tomato width/area (PWA). APC-cKit positive cells were detected using red laser, bandpass 660/20. Ckit negative gates were established using a sample of cells incubated with APC-Rat IgG instead of APC-cKit. Four-way sorting was performed using 100 µm nozzle and threshold was kept in the range of 2500–3000 events/second as PDD cells were fragile and sensitive to high pressures that can develop during sorting. Characterization and purity assessment of sorted cells was done by measurement of DNA content of the isolated populations (before and after sorting) and staining for specific markers of whole cells and nuclear spreads, as described below. Purity of populations was calculated by counting n ≥ 100 stained cells or nuclear spreads using ImageJ software and by DNA content estimated by propidium iodide after sorting. The PDD population consists ~6% cells in diakinesis. For studies of the diakinesis stage, we isolated cells that appeared in high FSC area at the gate of PDD. We achieved ~50% enrichment of these rare cells.

## DNA content

During sorting, DNA content estimation was performed by incubating cells (directly after dead cell removal procedure) with 10 μg/ml Hoechst for 30 min at 32°C in the dark. In sorted and isolated populations, DNA content was estimated by PI staining. Cells were washed with cold PBS and dispersed in 250 μl of PBS. Cells were fixed by adding 0.75 ml of 100% high-purity ethanol dropwise while slowly vortexing to a final volume of 75% ethanol. Fixed cells were washed twice with 1 ml cold PBS and spun down at 500 x g for 10 min at 4°C after each wash. Cells were resuspended in 0.2 ml "PI-Mix" (Materials) and incubated for 15–30 min in the dark before analysis. tdTomato fluorescence disappeared after ethanol fixation, allowing us to detect PI wavelength in tdTomato/PE channel (S1D Fig).

## Immunofluorescence staining of sorted cells for identification of early meiotic populations

Isolated cells were attached to poly L-lysine pretreated slides (1 mg/ml, 10 min) using cytospin (500 rpm) 3 min, low velocity to prevent PD cell damage). Cells were fixed by dipping the slides in methanol at -20°C for 30 sec. Slides were washed three times with PBS (3 min), then washed for 10 min in 0.1% Triton X-100 in PBS and blocked with 3% BSA, 5% donkey serum and 0.1% Triton X-100 in PBS for 1 hr. Primary antibody solution was prepared in 0.1% Triton X-100, 1%BSA in PBS, rabbit anti PLZF 1:100, mouse anti DMRT1 1:100, and slides were incubated overnight at 4°C. After overnight incubation, slides were washed 3 times for 10 min in PBS 0.1% Triton X-100 and incubated with secondary antibodies: FITC conjugated anti-mouse 1:1000 and Cy5 conjugated anti-rabbit 1:500 in PBS with 1% BSA for 2 hr in dark at room temperature. Following 3 washings for 3 min each in PBS, slides were covered with mounting solution containing DAPI and imaged with an Olympus FV3000 confocal microscope. The prevalence of PLZF and DMRT1 expressing cells was evaluated after counting at least 100 cells using ImageJ (S1E-H Fig).

## Nuclear spreads preparation and immunofluorescence staining

Nuclear spreads were prepared as described [81] with modifications as follows: Cells were collected into Eppendorf tubes and washed once with PBS (350 x g, 5 min) and all liquid was removed. Cells were dispersed in 150 μl of "hypotonic solution" (Materials) containing 0.5 mM PMSF, protease inhibitor cocktail X1 and 0.5 mM DTT, then 150 μl of PBS was added and tubes were incubated in rt for 8 min with periodic flickering. After incubation, 1.2 ml PBS was added and cells were spun for 5 min at 1000 x g. Most of the liquid was removed, 8 μl of the left liquid was moved to a new Eppendorf tube and 16 μl of sucrose 100 mM, pH=8.2 was added. Ten to twenty thousand nuclei in 6 μl of the above solution were spread dropwise on framed 2 x 1.5 cm areas on a slide covered with 42 μl freshly prepared and filtered solution of 1% PFA, 0.15% Triton X-100, 1 mM sodium borate, pH=9.2. Slides were put in closed humid chambers overnight. The next day, slides were air dried for approximately 1 hr, washed twice in 0.4% Photo-flo in water (2 min), then washed thrice in water (1 min each) and stored in PBS at 4°C until immunostaining. For immunostaining, slides were washed for 5 min in 0.5% Triton X-100 in PBS, 5 min in 0.1% Triton X-100 in PBS and blocked in filtered 2% BSA, 0.1% Triton X-100, 10% donkey serum in PBS for 1 hr. Primary antibodies were prepared in filtered 2% BSA, 0.1% triton X-100 and 10% donkey serum: rabbit anti-SCP3 1:750, mouse anti-γH2AX 1:3000, rabbit H3K9me3 1:500, rabbit HP1α 1:250, human CREST serum 1:20, rabbit LDHC 1:50, rabbit LDHA 1:100, then slides were incubated overnight at 4°C. The next day, slides were washed thrice for 5 min in 0.1% Triton X-100 in PBS, 5 min in PBS. Secondary antibodies were diluted in PBS with 1% BSA at the following dilutions: FITC-conjugated anti-mouse 1:500, CY5 conjugated anti-rabbit 1:500, Alexa 647 conjugated anti-human 1:100 and slides were incubated for 2 hr in dark at rt then washed thrice for 5 min PBS, followed with mounting solution containing Hoechst 1:250 (Vector) or Prolong Glass antifade with NucBlue. After staining, chromosomes were observed under Nikon Eclipse Ti microscope at X100 (Figs 4B, 4C, 8A and S1E-H) or LSM 980 Zeiss Airyscan2 at X63 magnification (Figs 3A, 3B, 3C, 4B-L, 5B-I, 5L-M, S2B-D, and S4A-H). Evaluation of immunostained areas and their intensity were performed using NIS elements or Zeiss 980 analyzer software. Identification of pachytene and diplotene cells was done by evaluation of the presence of synapsed and asynapsed regions of the bivalents based on SCP3 staining.

High-resolution images of chromosomes and centromeres presented in Fig 4D were acquired using Zeiss Lattice SIM3 X100 oil. The high-resolution images of centromeres shown in Fig 5J and 5K were acquired using Nikon N-SIM at X100 silicon.

Nuclear spreads of LDHC KO mice were prepared using non-enzymatically isolated testis cells [82]. After immunostaining with SCP3 and LDHC antibodies, chromosomes were observed using the Fluoview FV1000 laser scanning confocal microscope from Olympus America (Fig 3E). Additionally, these spreads were stained with CREST and HP1α (S4F and S4G Fig), and the staining areas were quantified using the LSM 980 Zeiss Airyscan2.

### Fresh cell nuclear spreads

Cells were isolated as described above, and after dead cells removal adhered to slides using cytospin and fixed by dipping in methanol at -20°C for 10 sec. Using this procedure, chromosomes could be visualized in whole cells. In addition, some PD cells released chromosomes, which could be analyzed using different antibodies for immunostaining. Slides were washed for 5 min in PBS and blocked in filtered 3% BSA, 0.1% Triton X-100, and 5% donkey serum in PBS for 1 hr. Primary antibodies were diluted in blocking solution: mouse anti-SCP3 1:200, LDHC 1:50, LDHA 1:100, slides were incubated overnight at 4°C with antibodies. The next day, slides were washed thrice for 10 min in 0.1% Triton X-100 in PBS, 5 min in PBS. Secondary antibodies were diluted in PBS with 1% BSA: FITC conjugated anti-mouse 1:500, CY5 conjugated anti-rabbit or anti-goat 1:500, and slides were incubated for 2 hr in dark at rt then washed three times for 5 min in PBS. Mounting was performed using ProLong Glass Antifade and, after 48 hours curation, pictures were taken with LSM 980 Zeiss Airyscan confocal microscope at X 63 magnification (Figs 3A, 3C, S2B, and S2D). Stimulated Emission Depletion Microscopy (STED) samples were prepared as described above, using Abberior STAR RED and Abberior STAR ORANGE secondary antibodies. Images were taken with an Abberior Instruments Facility LINE microscope equipped with an inverted IX83 microscope (Olympus), a 60x oil objective (UPlanXApo 60x/1.42 oil, Olympus), using pulsed excitation lasers at 561 nm and 640 nm and a pulsed STED laser operating at 775 nme. All acquisition operations were controlled by Lightbox Software. STED images presented in the (Fig 3D and 3F). were only adjusted in brightness and contrast on raw data using Fiji software 6.

### Histological sections

Testes were removed, fixed in formalin (for immunofluorescence and immunohistochemistry) or Bouin's fixative (for hematoxylin/eosin, PAS, or cleaved caspase 3 stains) overnight and embedded in paraffin (S6 Fig). Histological sections were immunostained with mouse anti-histone H3 phospho-serine 10 (PHH3) 1:400 and rabbit anti MAD1 1:100 after antigen retrieval in citrate buffered and blocking in filtered 2% BSA, 0.1% Triton X-100, and 10% donkey serum in PBS for 1 hr (Fig 8C and 8E).

### Culturing of isolated cell populations

LZ and PD cells were maintained in "culturing medium" for LZ and PDD ("Materials"), at 32°C as previously described [83]. L –or D-2HG-octyl, final concentration 0.3 mM or 0.6 mM, for 24–48 hours or 10 min incubations, respectively, was added dissolved in DMSO, 1.5 µl/ml media, sodium oxamate, final concentration 24 mM, was added dissolved in water, 54 µl/ml media. Untreated cells received equivalent volumes of solvents. When LZ cells were cultured for more than 48 hours, half of the media was replaced with fresh media containing drug or vehicle every two days.

### RNA-seq

Total RNAs from each sorted population from n = 3 independent experiments (Figs 1C and S1I) were isolated using a Qiagen mini kit according to the manufacturer's instructions. PolyA mRNAs were captured using magnetic oligo dT beads followed by fragmentation and cDNA synthesis. Libraries were constructed from fragmented DNA while performing end repair, A-base addition, adapter ligation and PCR amplification steps with SPRI beads cleanup in between steps. Indexed

samples were pooled and sequenced in an Illumina HiSeq 2500 machine in a single read mode. Median sequencing depth was ~ 18 million reads per sample. Sequencing depth was homogenous across samples in the experiment. Performed by G-INCPM (Weizmann Institute of Science).

To evaluate changes in the transcriptome of cultured cells under different treatments (Figs 1G, 1H, 7A, and 7B), RNAs were isolated using the method described above. The RNA quality was assessed with the TapeStation, employing the RNA ScreenTape kit from Agilent Technologies, and quantified using the Qubit apparatus with the Qubit DNA HS Assay kit from Invitrogen. Libraries were prepared from RNA samples using the KAPA Stranded mRNA-Seq Kit. These libraries were barcoded and pooled for multiplex sequencing (1.5 pM total, including 1.5% PhiX control library). The pooled DNA was loaded onto the NextSeq 500 High Output v2 kit (75 cycles) cartridge from Illumina and sequenced on the Illumina NextSeq 500 System under sequencing conditions of 75 cycles, single-read. The library preparation and sequencing processes were carried out at the Core Facility of the Hebrew University Faculty of Medicine.

Raw reads were processed for quality trimming and adaptors removal using fastx_toolkit v0.0.14 and cutadapt v2.10 [84]. The processed reads were aligned to the mouse transcriptome and genome version GRCh39 with annotations from Ensembl release 106 using TopHat v2.1.1 [85]. Counts per gesune quantification was done with htseq-count v2.01 [86]. Normalization and differential expression analysis were done with the DESeq2 package v 1.36.0 [87]. The DESeq2 likelihood ratio test (LRT) was used to identify genes that show change in transcript levels in any of the 3 treatments in the same test. This was done by using the full model design = ~Condition and the reduced model: reduced = ~ 1. Genes with a sum of counts less than 10 over all samples were filtered out. A matrix of normalized counts per sample of the genes with p-value < 0.05 were subjected to K-mean clustering using R. We generated clustering solutions of k = 2–10. Visual inspection led to the selection of k = 9 for PDD cells and k = 6 for RS as the optimal clustering solution. Gene clusters were plotted in heat maps (normalized base-mean counts per sample, after z-scoring per gene are displayed with sorting of genes by distance from cluster centroids)

### Tracing with UC13-L-lactate, UC13-D-glucose and UC13-glutamine

Cells were sorted into PBS with 5% dialyzed serum. LZ and PDD cells (0.5-1 million cells were incubated in 200 µl of K salt solution supplemented with 10% dialyzed serum, 2 mM glutamine and with 5 mM UC13 L-lactate, UC13 D-glucose or 2 mM UC13 glutamine in Eppendorf tubes for 120 min at 32°C. At the end of the incubation, cells were washed twice in PBS and extracted in methanol-acetonitrile (5:3). Extracted metabolites were separated on a SeQuant ZIC- pHILIC column (2.1 × 150 mm, 5 µm bead size, Merck Millipore). Flow rate was set to 0.2 ml/min, column compartment temperature was set to 30°C, and the autosampler tray was maintained at 4°C. Mobile phase A consisted of 20 mM ammonium carbonate with 0.01% (v/v) ammonium hydroxide. Mobile Phase B was 100% acetonitrile. The mobile phase linear gradient (%B) was as follows: 0 min 80%, 15 min 20%, 15.1 min 80%, and 23 min 80%. A mobile phase was introduced to the Thermo Q-Exactive mass spectrometer with an electrospray ionization source working in polarity switching mode. Metabolites were analyzed using full-scan method in the range 72–1,080 m/z and with a resolution of 70,000. Ionization source parameters were as following: sheath gas 25 units, auxiliary gas 3 units, spray voltage 3.3 and 3.8 kV in negative and positive ionization mode respectively, capillary temperature 325 °C, S-lens RF level 65, auxiliary gas temperature 200 °C. Positions of metabolites in the chromatogram were identified by corresponding pure chemical standards. Data was analyzed using the MAVEN software suite [88]. Relative metabolite levels were quantified from peak areas and normalized to unit mean for each peak after correction for protein content (Figs 2A, 2B, 4A, and 6A).

### Identification of 2HG enantiomers

Cell extracts were prepared using 50% MeOH + 30% Acetonitrile + 20% PBS mixture. Evaporated standards and samples were derivatized with TSPC by chiral derivatization approach and evaporated. For analysis, samples were re-dissolved in 100 µl of 30%-aqueous acetonitrile containing phthalic acid as internal standard (10 µM), centrifuged twice at 21,000 x g

for 5 min to remove insoluble material. The LC–MS/MS instrument consisted of an Acquity I-class UPLC system (Waters) and Xevo TQ-S triple quadrupole mass spectrometer (Waters) equipped with an electrospray ion source and operated in positive ion mode. MassLynx and TargetLynx software (version 4.1, Waters) were applied for data acquisition and analysis. Chromatographic separation was done on a 100 mm × 2.1 mm internal diameter, 1.7 μm UPLC BEH C8 column equipped with 50 mm × 2.1 mm internal diameter, 1.7 μm UPLC BEH C8 pre-column (both Waters Acquity) with mobile phases A (0.1% formic acid) and B (0.1% formic acid in 95%-aqueous acetonitrile) at a flow rate of 0.25 ml/min and column temperature 25°C. A gradient was used as follows: 0–1 min a linear increase from 30 to 32% B, then held at 32% B till 9.5 min, increase to 80% B during 1.5 min, then back to 30% B in 0.5 min, and equilibration at 30% B for 2.5 min, providing total run time of 13 min. Samples kept at 8°C were automatically injected in a volume of 3 μl. MS parameters (negative polarity): capillary voltage - 2.4 kV, source temperature -120°C. MRM transitions (collision energy, eV): 448.0 > 155.0 (29) and 448.0 > 318.0 (13) for derivatized 2HG, 446.0 > 155.0 (29) and 165.0 > 77.0 (15), 165.0 > 121.0 (10) for internal standards. Performed by Life Science Core Facilities, Weizmann Institute of Science (S2A Fig).

## Quantitative PCR of minor and major satellite transcripts

Cells were harvested with Trizol for RNA extraction. RNA was prepared using miRNeasy mini kit. Reverse transcription was carried out using SuperScript III First- strand Synthesis System The quantification of PCR products was analyzed with SYBR green using ABI PRISM 7900 Sequence Detection system software (Applied Biosystems,). The primer sequences were MajSAT 5′- GGCGAGAAAACTGAAAATCACG-3′,5′- CTTGCCATATTCCACGTCCT-3′; MinSAT5′-TGGAAACGGGATTTGTAGA-3′, 5′-CGGTTTCCAACATATGTGTTTT-3′. Results were normalized to Tox4 5'- TCCCG GAGGAAATGACAATTACC-3', 5'-GTGAGGGATCAGAGTCCAAGG-3' and Brd2 5'- AATGGCTTCTGTACCAGCTTTAC-3', 5'- CTGGCTTTTTGGGATTGGACA-3' [73] (Fig 4O and 4P) and (S3G and S3H Fig).

## HP1α and histone modifications

PDD cells were isolated and incubated for 48 hours with vehicle, 24 mM oxamate or oxamate with 0.3 mM L-2HG in duplicates. Cells were lysed in RIPA containing 1X protease inhibitors. Lysates were frozen at -20°C. Following thawing, they were sonicated for 3 x 30 sec and cleared by centrifugation at 14,000 x g for 15 min at 4°C. An equal amount of protein was extracted from 0.4 million cells and loaded onto a 12% acrylamide gel. HP1α and HK1 were probed with their respective antibodies, HK1 was used to normalize HP1α amounts, (Fig 4M). Histone lysine modifications were measured following the extraction of histones using the ab113476 histone extraction kit from Abcam. Multiplex colorimetric assay kit ab185910 with 25 ng protein per well (Fig 7C), and immunoblots were employed to assess semiquantitative changes in histone methylation (Fig 7D and 7F)

## Sample preparation for the whole cell proteomic mass spectrometry (MS) analysis

Five to six replicates of 0.3 million LZ and 0.15 million PDD cells were isolated from n = 5 mice and collected via sorting into PBS buffer. Cells were lysed with 5% SDS, 50 mM Tris-HCl pH 8.0, sonicated, and centrifuged at 14000 x g for 15 min at room temperature (Fig 2D and 2E).

## Isolation of chromatin-bound proteins for proteomic MS analysis

The isolation of chromatin–bound proteins was performed with the modification of a previously described method [37,38,89]. The composition of the solutions used is listed in "Materials". Briefly, 6 million LZ cells and 12 million PDD cells were isolated from n = 6 mice. Cells were homogenized in homogenization buffer followed by a 1000 X g centrifugation for 5 min to isolate nuclei. Low salt extraction (LSE) was performed by vigorous mixing nuclei in 1 ml LSE- buffer and an 100,000 x g, 4°C centrifugation for 30 min. The supernatant was discarded and the precipitates containing chromatin were

washed twice with 1 ml washing buffer (WB), mixing by vortex and centrifugation as described above. Chromatin-bound proteins were extracted with high salt extraction (HSE) buffer added by extensive sonication. After centrifugation, as described above, HSEs were collected for MS. (Fig 3G).

**LC MS/MS analysis of proteins**

The soluble extracts were supplemented with 10 mM dithiothreitol and incubated for 10 min. at 80°C. The proteins were alkylated by the addition of 55 mM iodoacetamide and incubation for 30 min at room temperature in the dark. Removal of SDS followed by digestion with sequencing grade modified trypsin (Promega Corp., Madison, WS) were performed using the S-Trap microspin column kit as specified by the manufacturer (Protifi, LLC, Huntington, NY, The tryptic peptides were desalted on C18 Stage tips [90]. A total of 0.3 μg of peptides (determined by Absorbance at 280 nm) from each sample were injected into the mass spectrometer.

MS analysis was performed using a Q Exactive Plus mass spectrometer (Thermo Fisher Scientific) coupled on-line to a nanoflow UHPLC instrument (Ultimate 3000 Dionex, Thermo Fisher Scientific). Eluted peptides were separated over a 120 min gradient run at a flow rate of 0.15 μl/min (during the separation phase) on a reverse phase 25-cm, C18 column (75 μm ID, 2 μm, 100Å, Thermo PepMap RSLC from Thermo Scientific). The survey scans (380–2,000 m/z, target value 3E6 charges, maximum ion injection times 50 ms) were acquired and followed by higher energy collisional dissociation (HCD) based fragmentation (normalized collision energy 25). A resolution of 70,000 was used for survey scans and up to 15 dynamically chosen most abundant precursor ions were fragmented (isolation window 1.8 m/z). The MS/MS scans were acquired at a resolution of 17,500 (target value 5E4 charges, maximum ion injection times 57 ms). Dynamic exclusion was 60 sec.

Mass spectra data were processed using the MaxQuant computational platform, version 1.6.17.0. Peak lists were searched against the mouse Uniprot FASTA sequence database of reviewed entries from Mar. 4, 2021, containing 36,759 entries. The search included cysteine carbamido-methylation as a fixed modification and oxidation of methionine and N-terminal acetylation as variable modifications. Peptides with minimum of seven amino-acid length were considered and the required FDR was set to 1% at the peptide and protein level. Protein identification required at least two unique or razor peptides per protein group. Relative protein quantification in MaxQuant was performed using the label free quantification (LFQ) algorithm [91]. LFQ in MaxQuant uses only common peptides for pairwise ratio determination for each protein and calculates a median ratio to protect against outliers. It then determines all pair-wise protein ratios and requires a minimal number of two peptide ratios for a given protein ratio to be considered valid.

Statistical analysis (n = 6) was performed using the Perseus statistical package [92]. The Perseus computational platform for comprehensive analysis of proteomics data. Only those proteins for which at least 3 valid LFQ values were obtained in at least one sample group were accepted for statistical analysis by Volcano plot (t-test, $p < 0.05$). After application of this filter, a random value was substituted for proteins for which LFQ could not be determined ("Imputation" function of Perseus). The imputed values were in the range of 10% of the median value of all the proteins in the sample and allowed calculation of p-values, (Figs 2D, 2E and 3G).

**Measurement of chromatin-bound LDHC enzymatic activity**

A total of 8 million PDD cells were isolated from $n = 5$ mice and divided equally into two tubes. Chromatin isolation was performed as detailed in the section "**Isolation of chromatin-bound proteins for proteomic MS analysis.**" Following isolation, chromatin was washed twice with WB buffer and used for enzymatic activity assays of chromatin-bound LDHC.

Enzymatic reactions were carried out using either 5 mM UC13- αKG or 1 mM UC13-pyruvate as substrates [12]. The reaction mixture consisted of 50 mM Tris-HCl (pH 7.4), 0.5 mM NADH in a total volume of 50 μl. Reactions were incubated for 12 hours at 34 °C. Negative control samples included all reaction components except chromatin. At the end of the incubation, chromatin was pelleted by centrifugation at 100,000 × g for 30 minutes at 4 °C. Subsequently, 20 μl of the

supernatant was extracted with 80 μl of methanol:acetonitrile (5:3). The reaction products, UC13-L-2HG and UC13-L-lactate, were analyzed by LC-MS as described in the section "Tracing with U-¹³C-L-lactate, U-¹³C-D-glucose, and U-¹³C-glutamine" (S2E and S2F Fig). Calibration curves were prepared by injecting known concentrations of L-2HG and L-lactate into the LC-MS. These curves enabled the calculation of LDHC activity, expressed as nanomoles of αKG converted to L-2HG and nanomoles of pyruvate converted to L-lactate, respectively.

## Statistical analysis

All experiments were repeated independently at least twice with similar results. For normal variable evaluations, data was shown as the mean value of replicates with their respective standard errors (mean ± SE); for non-parametric variable evaluations, data was shown as median with range as indicated in the figure legends. Student's t test, Mann Whitney, Friedman, Bonferroni and Tukey tests were performed as appropriate. P-values were calculated by using Graphpad Prism 9 software (San Diego, CA, USA). Levels of significance are indicated as $*p < 0.05$; $**p < 0.01$; $***p < 0.001$; $****p < 0.0001$. Statistical details for all experiments can be found in the respective figure legends. All tests were two tailed unless indicated otherwise. No statistical methods were used to predetermine sample size. Unless otherwise stated, experiments were not randomized, and investigators were not blinded to allocation during experiments and outcome assessment.

## Materials

(2S)-Octyl-α-hydroxyglutarate (HY-103641) MedChemExpress dissolved in DMSO

(R)-Octyl-α-hydroxyglutarate (SML2200) Merck dissolved in DMSO

Bovine serum albumin fatty acids free (A8806-5G) Sigma

Clarity Western ECL Substrate (170 5061) Bio-Rad

Collagenase (C5138) Sigma dissolved in DMEM/F12 medium 8 mg/ml

Collecting Buffer (during sorting) for experiments in which cells were cultured consisted of DMEM/F12 supplemented with 10% dialyzed serum, 3mM L-lactate, 3mM pyruvate and antibiotic-antimycotic solution.

Collecting Buffer (during sorting) for gene expression consisted of PBS with 2% albumin Collecting Buffer (during sorting) for tracing studies consisted of PBS with 5% dialyzed FBS.

Culturing medium (LZ and PD): MEM eagle (Ref # 01–040-1A from BI) supplemented with 2mM glutamine, 5mM sodium lactate, 5mM sodium pyruvate, 15mM HEPES, 1% antibiotic-antimycotic (Thermo Scientific, 15240096) v/v, 5% dialyzed serum.

Dead cells removal kit (130-090-101)

Miltenyi Dialyzed Serum (040111B)

Biological Industries DNAse (DN-25) Sigma dissolved in saline 7mg/ml FBS heat inactivated (04-127-1) Biological Industries

Flow Cytometry Size Calibration Kit (F13838) Thermo Fisher Scientific Gels for Western 4568094 Bio-Rad

Hoechst 33342 (H1399) Invitrogen, stock solution 1mg/ml.

Solutions for the extraction of chromatin-bound proteins for proteomic mass spectrometry analysis, and for measuring the enzymatic activity of chromatin-bound LDHC:

1. Homogenization solution - 0.32 M sucrose, protease inhibitors cocktail, 1mM PMSF and 10mM HEPES, pH 7.4.

2. Low salt extraction buffer (LSE)- 100mM KCl, 0.4mM EDTA, 0.1% Triton X-100, 10% glycerol, 1mM mercaptoethanol, protease inhibitors cocktail and 20mM TrisHCL, pH 7.5

3. Wash buffer (WB)- 1mM CaCl2, 1.5mM MgCl2, 10% glycerol and 10mM TrisHCl, pH 7.5.

4. High salt extraction (HSE)- 400mM KCL, 5mM MgCl, 2% Tween-20, 10% glycerol, 1mM mercaptoethanol, protease inhibitors cocktail and 20mM Hepes-KOH, pH 7.

Hypotonic solution (for spreads): 30mM tris-HCl, 100mM sucrose, 17mM sodium citrate, 5mM EDTA, PH = 8.3

K-salt solution: 142mM NaCl, 5.4mM KCl, 0.91mM NaH2PO4, 1.8mM CaCl2, 0.8mM MgSO4 and HEPES 5mM, pH 7.35 (*P/D cells are sensitive to Mg concentration. Higher than 0.8mM Mg can cause high death rate)

Medium DMEM/F12 01–170-1A Biological Industries miRNeasy mini kit (217004) Qiagen

Mounting solution for immunofluorescent staining ProLong Glass Antifade P36983 Invitrogen Photo-flo (501 0640) KODAK

PI-mix consists of PBS with (50µg/ml) Propidium Iodide and 50µg/ml RNAse-A

Poly L-lysine (p4707) Sigma.

Propidium Iodide- PI (P1304MP) Invitrogen, stock solution 250 µg/ml.

Protein assay- (5000006) Bio-Rad

RNeasy micro kit (74004) Qiagen RNeasy mini kit (74104) Qiagen

Sodium L-lactate 13-C3 (CLM-1577-0.5) Cambridge isotopes laboratories.

Sodium oxamate (O2751) Sigma

The sorting buffer consisted of PBS with 25mM HEPES, 5% dialyzed FBS, 0.05mg DNAse, and 0.75mM MgCl2

Trans-Blot Turbo Transfer Pack-(1704157) Bio-Rad

Trypsin-EDTA (03-0521-1) Biological Industries

A list of antibodies used in this study is provided in the supplementary information (S4 Table).

## Supporting information

**S1 Fig. Stra8-Tom mice allow isolation of highly purified populations of germ cells along their differentiation process. A.** tdTomato fluorescence (red) and immunostaining for Sox9 (green), demonstrating germ cells specificity of tdTomato expression, frozen section of Stra8-Tom mouse. **B**. 3D reconstruction of a whole mount preparation of seminiferous tubule, demonstrating decreasing tomato fluorescence towards the lumen. **C**. Stra8-Tom mice were produced by mating Stra8-icre males with CAG-LSL-tdTomato females (left). Stra8 icre-RosaYFP males were produced by mating Stra8 icre males with Rosa-LSL-YFP females (right). Shown are frozen sections counterstained with DAPI. Note that while the Rosa locus drives expression throughout the sperm lineage, expression driven by the CAG promoter gradually diminishes. The luminal fluorescence represents autofluorescence of sperm tails. **D**. DNA content was measured by adding Hoechst to testis cell suspensions followed with FACS analysis based on tdTomato intensity (before sorting) vs. Hoechst. Alternatively, cell populations were sorted as described, ethanol- fixed and stained with propidium iodide. The purity of cell populations based on DNA content after sorting was estimated as following: Undiff.Spg-100% 2C, Diff.Spg, PreL and LZ – mixed population of cells between 2C (incomplete replication) and 4C (complete replication), PDD – mostly 4C cells with slight contamination with 2C and about 14% 1C, Sec.Sp85% 2C and 15% 1C, RS-100% 1C. **E**. Isolated populations were attached to slides by cytospin centrifugation and immunostained for PLZF and DMRT1 (markers of spermatogonia). 90% of undifferentiated spermatogonia (ckit negative) were strongly positive for both markers, 90% of differentiating spermatogonia (ckit positive) were weakly positive for both markers and absent in all other populations. PLZF expression was 4.4, 4.8 and 10 fold higher in Undiff.Spg versus Diff.Spg in 3 respective experiments. GFR-α (another Undiff.Spg marker) expression was 8.7, 10 and 14-fold higher in Undiff.Spg. **F.** PreL cells were isolated and nuclear spreads were stained with antibodies for SCP3 and γH2AX. γH2AX staining was absent (upper panel) or present (lower panel). **G**. LZ cells were isolated and nuclear spreads were stained with antibodies for SCP3 and γH2AX. 44% of cells do not express γH2AX (PreL kind of cells, first row). 56% are a mixture of cells expressing different levels of γH2AX and different levels of synaptonemal complex formation (estimated by counting of 100 cells). **H.** PDD cells were isolated and nuclear spreads were stained with antibodies for SCP3 and γH2AX. The average composition of this population is: 40% pachytene, 40% diplotene, 6% diakinesis and 14% round spermatids. Scale bar **E-H**: 10 µm. **I**. mRNA was extracted from the freshly isolated indicated

populations of 3 mice. Shown are the numbers of genes which are differentially expressed (at least 2-fold difference, FDR).
(TIF)

**S2 Fig. LDHC generates L-2HG, is expressed in PDD cells and can be found in the nucleus. A.** L- and D-2HG enantiomer content was measured in the indicated populations using chiral derivatization. Results of two independent experiments are shown. Calculation of 2HG concentrations was performed using the volumes of cells in S2 Table. D-2HG enantiomer was detected only in Spg cells at negligible concentrations of 0.6 and 0.4 µM. **B-D**: Verification of LDHC staining along chromosomes and in centromeres. **B.** Freshly isolated testicular cells fixated for 10 sec in -20° methanol were stained with goat Abcam ab3966 LDHC antibody (and not rabbit Proteintech 19981-A) and co-stained with SCP3, compare with Fig 3A. **C.** Spreads prepared using Papanikos et al. [81], protocol described in methods 29. LDHC was stained with rabbit Proteintech 19981-A and co-stained with SCP3. Strong staining of centromeres with LDHC can be noted. Scale bar 10 µm. **D**: LDHA is not localized on chromosomes. Confocal microscope image of a diplotene cell stained with LDHA antibody, scale bar 10 µm, compare with Fig 3A. **E-F.** Catalytic activity of chromatin-bound LDHC was measured as described in the Methods. LC-MS was used to quantify metabolite levels. The M+5 peak area of L-2HG **(E)** and the M+3 peak area of lactate **(F)** were measured in the presence or absence of chromatin and either U-$^{13}$C-αKG **(E)** or U-$^{13}$C-pyruvate **(F)**, respectively (n=3 technical replicates).
(TIF)

**S3 Fig. The effect of oxamate on chromocenters and satellite transcripts. A.**Changes in centromeres composition during prophase 1. Pachytene, diplotene and diakinesis cells were stained with SCP3 (green) and CENPA, CENPC, CENPN and CREST antibodies (red). Note gradual appearance of CENPC and CENPN as compared with CENPA. **B.** Chromocenters demarcated by HP1α antibody are more prominent in diplotene cells as compared to pachytene cells. **C-H** PDD cells were isolated from two groups: mice injected once with oxamate (1.3g/kg) and sacrificed after 24 hours (Ox) and mice injected with vehicle (Ctr). **C-F.** Chromocenters of diplotene cells (n=60, analyzed with a t-test) stained with H3K9me3 were evaluated for the following parameters: area **(C)**, staining intensity **(D)**, DAPI intensity **(E)**, and the ratio of H3K9me3 to DAPI intensities **(F)**. **G-H** qPCR performed for minor **(G)** or major **(H)** satellite RNA. Data represent mean±SE from 3 experiments, t-test.
(TIF)

**S4 Fig. The absence of any discernible impact from low 2HG level on various chromosome parameters, as well as on centromeres and chromocenters in LDHC-KO mice. A-C**. The PDD population was isolated and cultured in medium with vehicle, 24 mM oxamate, and 24 mM oxamate supplemented with 0.3 mM octyl-L/D 2HG for 24 hours. A. After incubation, cells were attached to slides using cytospin and fixated for 10 seconds in -20°C methanol. SCP3 staining was performed to identify diplotene cells, and nuclei area was measured based on DAPI staining (70–100 nuclei in each group, one-way ANOVA). **B-C**. Nuclear spreads were prepared after incubation, and pachytene chromosomes were identified using SCP3 staining. **B.** Chromosome length was measured using ImageJ. Each dot represents the mean length of chromosomes of one cell, with 60 cells per group (Mann-Whitney). **C.** Width was measured using Zeiss software. Each dot represents the mean width of chromosomes of one cell, with 20–30 cells in each group (one-way ANOVA). **D-E** and **H**. PDD cells were isolated from two groups: mice injected once with oxamate (1.3g/kg) and sacrificed after 24 hours (Ox), and mice injected with vehicle (Ctr). **D.** Cells were stained with SCP3 and MLH1. The number of MLH1 foci was counted in 25 and 30 cells of the respective groups, with mean±SE calculated (t-test). **E.** A representative image of MLH1 staining is shown, with a scale bar of 10 µm. **F-G.** Cells were isolated, and nuclear spreads were prepared from WT and LDHC-KO mice. Cells were stained either with CREST and SCP3 to measure centromere area **(F)** or with HP1α and SCP3 to measure chromocenter area **(G)**. There were 20–25 diplotene cells in each group (t-test). **H.** Diplotene cells from control and

oxamate-injected mice were stained for CREST and SCP3, and centromere area was measured in 30 cells in each group (t-test).
(TIF)

**S5 Fig. L-2HG concentration regulates centromere separation in diakinesis cells. A-B.** The PDD population enriched for diakinesis cell content was cultured in medium with vehicle, 24 mM oxamate, and 24 mM oxamate supplemented with 0.3 mM octyl-L-2HG for 24 hours. Following incubation, nuclear spreads were prepared and stained with SCP3 and CREST. **A.** Representative images with white circles marking paired centromeres are shown, with a scale bar of 10 µm. **B.** The number of centromere pairs was counted in 20 cells, mean±SE, one-way ANOVA.
(TIF)

**S6 Fig. Lack of histological signs of toxicity after 7 days of treatment with oxamate.** Mice were treated with daily oxamate injections 1.3 g/kg or vehicle for 7 days (n = 6). Paraffin sections were stained with PAS for morphological evaluation or with cleaved caspase 3 for identification of apoptotic cells. Representative images one out of 6 mice, scale bar 100 µm.
(TIF)

**S1 Table. Cultured cells are not significantly contaminated with myoid cells.** Gene expression of LZ (6 days in culture) and PDD (48 hours)- normalized reads, mean ±SE of 3 samples. Data from GSE169014. Data were from freshly isolated (e.g., not cultured) cells from GSE162740.
(PDF)

**S2 Table. Cell size parameters of the different cell populations.** Cell volumes were measured using two independent methods – eclipse and diameter measurement – under the microscope, and there was a high level of agreement in results from the two methods. The volume ratio between the respective cells was calculated based on the average of the two methods.
(PDF)

**S3 Table. Summary of significant effects attributed to alterations in L-2HG levels on gene categories related to chromatin organization and remodeling.** Genes from clusters 3 and 9 of the PDD population and clusters 1 and 2 of the RS population were analyzed using the metadatabase GeneAnalytics software, with the pathway enrichment significance threshold set at FDR < 0.05.
(PDF)

**S4 Table. A list of antibodies.**
(DOCX)

**S1 Data. List of genes affected by modulation of L-2HG levels in clusters 3 and 9 of PDD cells.**
(XLSX)

**S2 Data. List of genes affected by modulation of L-2HG levels in clusters 1 and 2 of RS cells.**
(XLSX)

## Acknowledgments

We thank Prof. Norman Grover for help with some of the statistical analyses and Drs Ittai Ben- Porath, Yuval Dor, Yotam Drier and Yehudit Bergman for helpful discussions and comments on the manuscript. We thank Mohammad Jumaa for excellent histological assistance, Dr. Douglas Lutz, Getter Group BioMed and Dr. Abel Pereira da Graça, Carl Zeiss

Microscopy, GmbH, for help with high resolution microscopy analyses, Dr. William Breuer for his assistance in proteomics analysis and Dr. Yuval Nevo for assistance with bioinformatic analyses.

## Author contributions

**Conceptualization:** Eli Pikarsky.

**Data curation:** Nina Mayorek, Yousef Mansour, Nir Pillar, Guy Baziza Paz.

**Formal analysis:** Nina Mayorek, Miriam Schlossberg, Yousef Mansour, Nir Pillar, Eli Pikarsky.

**Funding acquisition:** Christopher B. Geyer, Eli Pikarsky.

**Investigation:** Nina Mayorek, Miriam Schlossberg, Yousef Mansour, Ilan Stein, Fatima Mushasha, Guy Baziza Paz, Eleonora Medvedev, Zakhariya Manevitch, Julia Menzel, Elina Aizenshtein, Boris Sarvin, Nikita Sarvin, Bryan A. Niedenberger.

**Methodology:** Nina Mayorek, Miriam Schlossberg, Yousef Mansour, Eleonora Medvedev, Zakhariya Manevitch, Julia Menzel, Elina Aizenshtein, Michael Klutstein.

**Project administration:** Ilan Stein, Eli Pikarsky.

**Resources:** Christopher B. Geyer, Tomer Shlomi, Michael Klutstein, Eli Pikarsky.

**Supervision:** Nina Mayorek, Christopher B. Geyer, Tomer Shlomi, Eli Pikarsky.

**Validation:** Nina Mayorek, Miriam Schlossberg, Yousef Mansour, Fatima Mushasha, Guy Baziza Paz, Boris Sarvin, Nikita Sarvin, Bryan A. Niedenberger.

**Visualization:** Nina Mayorek, Miriam Schlossberg, Yousef Mansour, Guy Baziza Paz.

**Writing – original draft:** Nina Mayorek, Christopher B. Geyer, Michael Klutstein, Eli Pikarsky.

**Writing – review & editing:** Nina Mayorek, Guy Baziza Paz, Erwin Goldberg, Christopher B. Geyer, Michael Klutstein, Eli Pikarsky.

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
