## [Decision Letter · Decision Letter 0]

PGENETICS-D-24-01544

2-hydroxyglutarate controls centromere and heterochromatin conformation and function in the male germline

PLOS Genetics

Dear Dr. pikarsky,

Thank you for submitting your manuscript to PLOS Genetics. After careful consideration, we feel that it has merit but does not fully meet PLOS Genetics's publication criteria as it currently stands. Therefore, we invite you to submit a revised version of the manuscript that addresses the points raised during the review process.

Please submit your revised manuscript within 30 days Apr 08 2025 11:59PM. If you will need more time than this to complete your revisions, please reply to this message or contact the journal office at plosgenetics@plos.org. Please include the following items when submitting your revised manuscript:

We look forward to receiving your revised manuscript.

Kind regards,

Nick Gilbert

Guest Editor

PLOS Genetics

Aleksandra Trifunovic

Section Editor

PLOS Genetics

Aimée Dudley

Editor-in-Chief

PLOS Genetics

Anne Goriely

Editor-in-Chief

PLOS Genetics

**Journal Requirements:**

At this stage, the following Authors/Authors require contributions: Nina Mayorek, Miriam Schlossberg, Yousef Mansour, Nir Pillar, Ilan Stein, Fatima Mushasha, Guy Baziza Paz, Eleonora Medvedev, Zakhariya Manevitch, Julia Menzel, Elina Aizenshtein, Boris Sarvin, Nikita Sarvin, Erwin Goldberg, Bryan A Niedenberger, Christopher B Geyer, Tomer Shlomi, Michael Klutstein, and Eli Pikarsky. Please ensure that the full contributions of each author are acknowledged in the "Add/Edit/Remove Authors" section of our submission form.

The list of CRediT author contributions may be found here: https://journals.plos.org/plosgenetics/s/authorship#loc-author-contributions

2) We note that your Supplementary Tables files are duplicated on your submission. Please remove any unnecessary files.

3) We ask that a manuscript source file is provided at Revision. Please upload your manuscript file as a .doc, .docx, .rtf or .tex. If you are providing a .tex file, please upload it under the item type u2018LaTeX Source Fileu2019 and leave your .pdf version as the item type u2018Manuscriptu2019.

4) We do not publish any copyright or trademark symbols that usually accompany proprietary names, eg ©,  ®, or TM  (e.g. next to drug or reagent names). Therefore please remove all instances of trademark/copyright symbols throughout the text, including:

- ® on page: 38

- TM on page: 30.

5) Please upload all main figures as separate Figure files in .tif or .eps format. For more information about how to convert and format your figure files please see our guidelines: 

6) We have noticed that you have uploaded Supporting Information files, but you have not included a list of legends. Please add a full list of legends for your Supporting Information files after the references list.

7) Some material included in your submission may be copyrighted. According to PLOSu2019s copyright policy, authors who use figures or other material (e.g., graphics, clipart, maps) from another author or copyright holder must demonstrate or obtain permission to publish this material under the Creative Commons Attribution 4.0 International (CC BY 4.0) License used by PLOS journals. Please closely review the details of PLOSu2019s copyright requirements here: PLOS Licenses and Copyright. If you need to request permissions from a copyright holder, you may use PLOS's Copyright Content Permission form.

Potential Copyright Issues:

i) Figures 5A, and S1C. Please confirm whether you drew the images / clip-art within the figure panels by hand. If you did not draw the images, please provide (a) a link to the source of the images or icons and their license / terms of use; or (b) written permission from the copyright holder to publish the images or icons under our CC BY 4.0 license. Alternatively, you may replace the images with open source alternatives. See these open source resources you may use to replace images / clip-art:

ii) We note that [Figure 9] is created through BioRender. Please confirm that you hold a Premium account and provide a pdf copy of the CC BY 4.0 Licence as provided by BioRender. For instructions on how to generate a CC BY 4.0 license for your figure, please see the guidelines here: https://help.biorender.com/hc/en-gb/articles/21282341238045-Publishing-in-open-access-resources. 

If you are using the free assets from BioRender, we are unable to publish these images as they are licenced under a stricter licence than CC BY 4.0. In this case we ask you to remove the BioRender images and replace them with open source alternatives.

See these open source resources you may use to replace images / clip-art:

- https://openclipart.org/

8) Please amend the description of manuscript to "'Manuscript" rather than " Cover Letter" in the online submission form.

**Reviewers' comments:**

Reviewer's Responses to Questions

Reviewer #1: This study identifies L-2HG as a critical regulator of centromere and heterochromatin organization in spermatogenesis, revealing a novel metabolic-chromatin link. It provides strong evidence that nuclear LDHC associates with meiotic chromosomes and plays a key enzymatic role. The finding that L-2HG depletion disrupts pericentric heterochromatin and centromere integrity is convincing. The proposed role of polymer-polymer phase separation (PPPS) in centromere structure adds mechanistic depth, with potential implications for male fertility and genome stability. The study is well-executed, well-written, and conceptually innovative. However, several concerns need to be addressed:

1. How does L-2HG regulate chromosome dynamics during meiosis? One possible mechanism is its direct binding to chromatin modifiers (Fig. 9). Additionally, the authors show in Fig. 7 that L-2HG manipulation induces global transcriptomic changes in genes controlling chromatin modifications, suggesting transcriptional regulation. Please clarify this point.

2. While Fig. 3 demonstrates nuclear localization of LDHC, it remains unclear whether nuclear LDHC has a metabolic function in L-2HG production. Additional experiments are needed to differentiate the roles of cytoplasmic and nuclear LDHC.

3. The use of oxamate to block L-2HG synthesis lacks specificity. Although this concern is partially mitigated by the rescue experiment using octyl-L-2HG, siRNA-mediated LDHC knockdown is feasible with the current culture system and would provide more direct evidence.

Reviewer #2: This study by Mayorek and colleagues examines the noncanonical functions of Lactate Dehydrogenase C (LDHC) in the mouse testis. Previous studies from Schatz and Segal in 1969 described how LDHC (then called LDHX) from mouse testis can synthesize 2HG from a-ketoglutarate. This function was rediscovered and elaborated on by the Rabinowitz group in 2016 – at which point the question was raised as to what endogenous function this LDHC-L2HG axis served in the mouse testis. Here the authors conduct a series of well-controlled experiments that reveal an important function for LDHC in centromere condensation and heterochromatin organization.

The manuscript is divided into two sections. The first section describes a novel genetic tool that enables sorting of testis cell-types. While this section could be better integrated with the latter 2/3 of the manuscript, the methodology appears sound and will likely be of broader interest. The second portion of the manuscript focuses on the function of LDHC on producing L-2HG in a subset of testis cell types, with a majority of efforts directed on the role of LDHC and L-2HG in regulating: (i) LDHC localization to chromatin, and (ii) maintaining centromeric heterochromatin. The experiments are well-controlled and convincing. Overall, the authors demonstrate that L-2HG promotes LDHC localization to chromatin. Moreover, their findings indicate that loss of L-2HG production disrupts heterochromatin density.

I think this is a landmark study – for too long, studies of both D-2HG and L-2HG have focused on disease phenotypes. Nearly all manuscripts focus on the links between the two 2HG enantiomers and either oncogenic mutations or human disease phenotypes. But an ever increasing body of evidence indicates that both D-2HG and L-2HG have important endogenous functions – bacteria generate D- and L-2HG, yeast generate D-2HG, Diptera – the second-largest order of animals on planet earth generate millimolar quantities of L-2HG during juvenile growth. Yet, studies of D- and L-2HG almost always focus on cancer metabolism.

My major criticism for the manuscript is that I'd encourage the authors to break out of the biomedical troupe that D- and L-2HG are oncometabolites and instead embrace the true importance of their findings. Theirs is the first conclusive evidence that L-2HG serves an important endogenous role during animal development. I’d strongly suggest that the authors restructure their introduction and discussion to claim this primacy. While they should feel free to discuss the oncogenic roles of these molecules, I’d note that they have the opportunity to restructure the manuscript in a manner that shifts the entire field towards thinking about the normal roles of the 2HGs. There’s plenty of evidence that these molecules play essential roles in development – you have the most concrete evidence to date.

I’d also insist that the authors clean up aspects of the manuscript related to L-2HG metabolism. Throughout the manuscript, the authors mention 2HG as if it’s one molecule. Be specific wherever possible – use L-2HG instead of 2HG unless the assay was ambiguous. Similarly, the authors conduct a series of elegant rescue experiments using octyl-L2HG. There are several instances where the authors note adding D- or L-2HG to cells – be specific, if these experiments involved octyl-2HG, please note this appropriately.

A few other notes for the text:

The intro mentions production of D-2HG in mammals due to oncogenic mutations in IDH. D-2HG is also produced under endogenous conditions due to the enzyme HOT and from serine biosynthesis.

The results sections focused on the new transgenic tools are conceptually divorced from the remainder of the manuscript. The authors should consider some edits to this text to setup the remainder of the results section.

Schatz and Segal described how LDHC from testis produces L-2HG in 1969. Sorry if I missed the reference, but this discovery should be given precedence over the Rabinowitz lab discovery in 2016.

The 13C labeling experiment, as written, implies that L-2HG is produced directly from lactate. I think the authors mean to imply that lactate is converted to pyruvate and fed into the TCA cycle to generate aKG, which is then used to synthesize L-2HG. Please state so explicitly.

The finding that oxamate, but not LDHC mutations, exhibit differences in centromere and heterochromatin compaction to be very interesting. The authors try to explain way this difference, but an alternative model is that LDHC unbound by L-2HG is important for the phenotype, and in the LDHC mutants, the absence L-2HG causes LDHC to promote changes in heterochomatin. The authors should raise this possibility.

Please cite appropriate references outside of the oncometabolite literature. The Schatz and Segal study listed above is essential. In addition, a PNAS study of L-2HG in Drosophila noted that loss of L-2HG results in enhanced position effect variegation phenotypes in the fly eye. This phenotype is known to result from spreading of pericentric heterochromatin and clearly supports the authors findings. While the fly study doesn’t describe a mechanistic basis of this phenomenon – the authors study provides an explanation for this finding in the fly.

Reviewer #3: This study investigated 2-HG in mouse spermatogenesis. Using biochemical approaches, they showed that 2-HG is most abundant in PDD cells (pachytene diplotene diakinesis spermatocytes) and round spermatids (RS). They further showed that 2-HG is produced by LDHC, a testis-specific LDH. LDHC localizes to the meiotic chromosomal axis and centromeres in spermatocytes. Acute treatment of PDD cells with oxamate, a pan-LDH inhibitor, causes enlargement of centromeres and chromosome centers in PDD cells in culture. Mice treated with oxamate exhibited the same defects in centromeres and chromocenters in spermatocytes. However, LDHC-deficient mice did not show similar defects. Treatment of round spermatids with oxamate also caused increase in chromocenter size. Transcriptomic profiling revealed increased minor and major satellite RNAs upon oxamate treatment. This study supports that conclusion that 2-HG modulates centromere and chromocenter conformation in male germ cells, specifically in meiotic spermatocytes and round spermatids.

This study is well done. The manuscript is well-written. Discussions are balanced. Alternative explanations (redundant enzymatic functions etc) were provided. I support its publication after addressing the following concerns.

Major concerns:

1) The title needs to be revised: “2-hydroxyglutarate controls centromere and heterochromatin conformation and function in the male germline”. This study didn’t examine the function of centromere and heterochromatin. Therefore, “function” needs to be deleted in the title. “control” is too strong. Suggested title: 2-hydroxyglutarate regulates centromere and heterochromatin conformation in the male germline.

2) Testis is naturally hypoxic. PDD cells and round spermatids are behind the blood-testis barrier and reside in a hypoxic environment. Could the high 2-HG in the testis help germ cells to adapt to hypoxia? A brief discussion would be useful to readers.

3) Localization of LDHC to the chromosomal axis (synaptonemal complex) and centromeres in PDD cells. This localization pattern of LDHC in PDD cells (meiotic prophase I cells) is novel and very interesting. However, the function of this localization remains unknown, because the LDHC-deficient males produced sperm and lacked meiotic defects. At least it is not required for meiotic progression. LDHC knockout males were infertile due to failure for the knockout sperm to fertilize oocytes.

4) Lines 103-104: This statement needs to be deleted: “These effects are necessary for the proper progression of meiosis and thus directly link 2HG to normal meiotic progression”. The data supporting this statement were weak, because the mice were only studied at 7 days. This might not be long enough to see the chromosome segregation defects. No meiotic defects were shown except for the MAD1 signal. In addition, activation of SAC would lead to apoptosis in spermatocytes. Increased apoptosis was not shown.

**Have all data underlying the figures and results presented in the manuscript been provided?**

Reviewer #1: Yes

Reviewer #2: Yes

Reviewer #3: Yes

PLOS authors have the option to publish the peer review history of their article (what does this mean? ). If published, this will include your full peer review and any attached files.

**Do you want your identity to be public for this peer review?** For information about this choice, including consent withdrawal, please see our Privacy Policy .

Reviewer #1: No

Reviewer #2: No

Reviewer #3: No

**Figure resubmission:**
---

## [Decision Letter · Decision Letter 1]

Dear Dr pikarsky,

We are pleased to inform you that your manuscript entitled "L-2-hydroxyglutarate regulates centromere and heterochromatin conformation in the male germline" has been editorially accepted for publication in PLOS Genetics. Congratulations!

Yours sincerely,

Nick Gilbert

Guest Editor

PLOS Genetics

Monica Colaiácovo

Section Editor

PLOS Genetics

Aimée Dudley

Editor-in-Chief

PLOS Genetics

Anne Goriely

Editor-in-Chief

PLOS Genetics

Comments from the reviewers (if applicable):

Reviewer's Responses to Questions

**Comments to the Authors:**

Reviewer #1: All comments are addressed.

Reviewer #2: The authors have addressed my concerns.

Reviewer #3: My concerns have been adequately addressed.

**Have all data underlying the figures and results presented in the manuscript been provided?**

Reviewer #1: Yes

Reviewer #2: Yes

Reviewer #3: Yes

PLOS authors have the option to publish the peer review history of their article (what does this mean? ). If published, this will include your full peer review and any attached files.

**Do you want your identity to be public for this peer review?** For information about this choice, including consent withdrawal, please see our Privacy Policy .

Reviewer #1: No

Reviewer #2: No

Reviewer #3: No

**Data Deposition**

http://datadryad.org/submit?journalID=pgenetics&manu=PGENETICS-D-24-01544R1

**Press Queries**

---

## [Editor Report · Acceptance letter]

PGENETICS-D-24-01544R1

L-2-hydroxyglutarate regulates centromere and heterochromatin conformation in the male germline

Dear Dr pikarsky,

We are pleased to inform you that your manuscript entitled "L-2-hydroxyglutarate regulates centromere and heterochromatin conformation in the male germline" has been formally accepted for publication in PLOS Genetics! Your manuscript is now with our production department and you will be notified of the publication date in due course.

With kind regards,

Zsofia Freund

PLOS Genetics

On behalf of:
